# Brain Dissection: fMRI-trained Networks Reveal Spatial Selectivity in the Processing of Natural Images

**Gabriel H. Sarch**    **Michael J. Tarr**    **Katerina Fragkiadaki**[*]    **Leila Wehbe**[*]

Carnegie Mellon University

{gsarch,mt01}@andrew.cmu.edu,katef@cs.cmu.edu,lwehbe@cmu.edu

[*]Equal advising

brain-dissection.github.io

## Abstract

The alignment between deep neural network (DNN) features and cortical responses currently provides the most accurate quantitative explanation for higher visual areas [1, 2, 3, 4]. At the same time, these model features have been critiqued as uninterpretable explanations, trading one black box (the human brain) for another (a neural network). In this paper, we train networks to directly predict, from scratch, brain responses to images from a large-scale dataset of natural scenes [5]. We then use "network dissection" [6], an explainable AI technique used for enhancing neural network interpretability by identifying and localizing the most significant features in images for individual units of a trained network, and which has been used to study category selectivity in the human brain [7]. We adapt this approach to create a hypothesis-neutral model that is then used to explore the tuning properties of specific visual regions beyond category selectivity, which we call "brain dissection". We use brain dissection to examine a range of ecologically important, intermediate properties, including depth, surface normals, curvature, and object relations across sub-regions of the parietal, lateral, and ventral visual streams, and scene-selective regions. Our findings reveal distinct preferences in brain regions for interpreting visual scenes, with ventro-lateral areas favoring closer and curvier features, medial and parietal areas opting for more varied and flatter 3D elements, and the parietal region uniquely preferring spatial relations. Scene-selective regions exhibit varied preferences, as the retrosplenial complex prefers distant and outdoor features, while the occipital and parahippocampal place areas favor proximity, verticality, and in the case of the OPA, indoor elements. Such findings show the potential of using explainable AI to uncover spatial feature selectivity across the visual cortex, contributing to a deeper, more fine-grained understanding of the functional characteristics of human visual cortex when viewing natural scenes.

## 1 Introduction

Humans possess an extraordinary ability to process and interpret visual information; however, our understanding of how such information is represented and processed within the brain remains incomplete. While advances in modeling visual cortex have been made through the use of abstract stimuli and the study of evoked neuronal responses, these efforts have primarily focused on low-level visual features [8] or high-level categorical selectivity [9]. In reality, the brain utilizes numerous ecologically-relevant intermediate features to perceive the world, such as 3D information, physical relationships, and object properties [10, 11].

37th Conference on Neural Information Processing Systems (NeurIPS 2023).

Characterizing high-level areas in the visual cortex presents challenges due to the circular problem of finding optimal stimuli. Although deductive [12, 13, 14, 15, 16] and inductive [17, 18, 19, 20] approaches have identified a wide range of functionally-defined brain regions, these methods are not scalable for exploring a wider range of brain areas and diverse features. To address these limitations, we leverage "network dissection" [6], a method that enables exploring the feature selectivity of neural network units at scale. Combined with brain response-optimized networks trained on fMRI data for subjects viewing natural images, this method offers a hypothesis-neutral approach to uncovering feature tuning properties across the visual system [7, 21]. While Khosla and Wehbe [7] demonstrate the effectiveness of network dissection for validating known category selectivities, here we report that this combined approach, *brain dissection*, can be extended to examine more granular features, such as depth, curvature, and object relations.

We apply brain dissection across the visual cortex to uncover functional differences between regions. In particular, the human visual system is believed to be organized into distinctive processing streams: the parietal, lateral, and ventral visual streams. While previous experimental and modeling work has made progress in characterizing each of these streams [22, 23, 24, 1, 25, 26], at a fine-grained level, their distinct roles in visual processing remain under-specified. Functional networks within these streams likewise remain functionally under-specified. For example, precise differences among the three components of the "scene network", the occipital place area (OPA), parahippocampal place area (PPA), and the retrosplenial complex (RSC) [27] remain unclear for naturalistic, large-scale viewing of scenes with prior work providing primarily broad functional characterizations in terms of the representation of scene layout [28, 29, 30], 3D features [20, 31, 32, 33], scene objects [15, 7], and navigational affordances [34, 35].

Utilizing a hypothesis-neutral approach with brain response optimization, we demonstrate feature selectivity differences across multiple visual streams and scene regions, encompassing a wide variety of highly specific feature properties. Importantly, our findings are based on subjects viewing natural images and do not require a separate experiment for each category or visual feature.

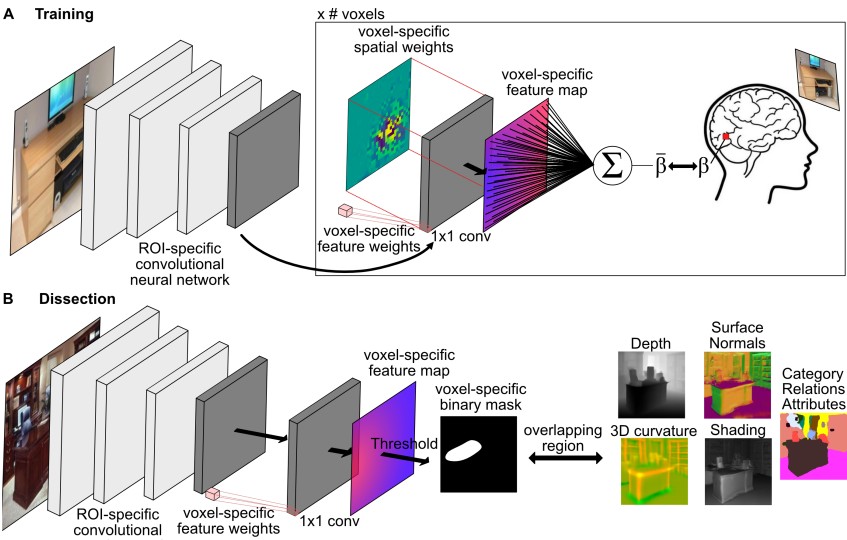

Figure 1: **A. fMRI Response-Optimized Training Procedure.** All voxels in an ROI share a convolutional backbone. Each voxel is assigned a set of learnable spatial and feature weights. The voxel weights are applied to the output features of the backbone network to predict the voxel response for an input image. Training minimizes the mean squared error between the predicted and actual voxel response of NSD images. **B. Dissection Procedure.** For each voxel in an ROI and test image, we obtain the voxel's selectivity binary mask by applying the voxel's feature weights to the output of the ROI-specific backbone and thresholding the resultant feature map. We examine the overlapping region between the mask and a measurement of interest (e.g., depth map).

## 2  Brain Dissection

Brain dissection provides a hypothesis-neutral approach for identifying the most significant image features for predicting the response of a specific voxel. The method trains a convolutional neural network tailored to predict voxel responses within a defined sub-region. By training a backbone network for this sub-region and incorporating a linear readout for each voxel, the network learns to extract image features crucial for predicting each voxel (Fig. 1A). After training, we "dissect" the network by inputting held-out images and extracting regions that the network considers most relevant for each voxel (Fig. 1B). Subsequently, we analyze properties of the voxel-selective regions within the images. Our training and dissection procedure closely follows Khosla and Wehbe [7].

### 2.1  fMRI Response-Optimized Network Training

We trained a set of neural network encoding models to predict fMRI voxel responses to NSD images (Fig. 1A). Visual cortex was divided into sub-regions based on the Human Connectome Project (HCP) atlas [36], with a separate convolutional neural network (CNN) backbone shared across all voxels within each sub-region. For each voxel in a sub-region, a linear readout was applied to the backbone output features to predict individual voxel responses to the images. The linear readout was factorized into voxel-specific spatial and feature dimensions, as described by Klindt et al. [37]. Spatial weights correspond to the most responsive locations of a voxel in an image, while feature weights represent the visual information to which a voxel is most responsive.

Given the activations of the last layer of the ROI-specific backbone network ($A \in \mathbb{R}^{C \times H \times W}$), the spatial weights for a voxel in the ROI ($w_s \in \mathbb{R}^{H \times W}$) were multiplied channel-wise to $A$. The feature weights ($w_f \in \mathbb{R}^C$) were applied to each spatial position of $A$. The output is a voxel-specific feature map, which was summed and added to a learned bias to obtain the predicted voxel response. The network was trained using mean-squared error with the true voxel response. We trained each network using NSD data from 8 subjects, including 68,400 training images, and 3,600 validation images. We show predictivity performance of our networks in Appendix Section S3.2.

While we use a CNN for all the analysis in the main paper, we also report results for the scene regions using a transformer architecture, and attention heads for the voxel-specific feature maps with consistent results in the Appendix Section S3.1.

### 2.2  Dissection Procedure

During dissection (see Fig. 1B), the spatial weights were discarded and only the feature weights were used to focus the analysis on visual selectivity for each voxel. For each voxel $k$, we obtained its corresponding ROI-specific backbone network $f_{roi}$ (with $k \in roi$) and voxel-specific feature weights $w_f^k \in \mathbb{R}^C$. For every input image $x$ in the evaluation dataset, we passed the image through $f_{roi}$ to obtain activations $A$. The feature weights ($w_f^k \in \mathbb{R}^C$) were then applied to each spatial position of $A$ using a 1x1 convolution. The output is a voxel-specific feature map $A_k \in \mathbb{R}^{H \times W}$. The top quantile level $T_k$ is determined such that $P(A_k > T_k) = 0.01$ over every spatial location of the activation map.

To compare a low-resolution voxel activation map to the input-resolution annotation $L_c$ for some concept or measure $c$ (e.g., $L_c$ could be a category mask, depth map, surface normal map, etc.), the activation map is scaled up to the mask resolution $S_k$ from $A_k$ using bilinear interpolation. $S_k$ is then thresholded into a binary segmentation: $M_k = S_k \geq T_k$, selecting all regions for which the activation exceeds the threshold $T_k$.

As in Bau et al. [6], to score each voxel $k$ for a discrete category concept $c$, we compute the data-set-wide intersection over union (IOU) score:

$$IOU_{k,c} = \frac{\sum_X |M_k(x) \cap L_c(x)|}{\sum_X |M_k(x) \cup L_c(x)|} \tag{1}$$

To score the selectivity of each voxel $k$ against a continuous concept $c$ (e.g., depth), we compute the median value within the region defined by $M_k$, and take the average median across all images as the selective value for that voxel:

$$V_k = \frac{\sum_X median(L_c(x)[M_k(x)])}{\sum |X|} \tag{2}$$

where $L_c(x)[M_k(x)]$ represents the values in $L_c(x)$ defined by the mask $M_k(x)$.

We exclude voxels that lack selectivity across all evaluation set images by comparing the distribution of L_c masked by M_k against a uniformly distributed (full image) mask. This exclusion used the Kolmogorov–Smirnov test, with a significance threshold of p>0.01.

While we present results using the Network Dissection interpretability method [6] throughout the main paper, we demonstrate in Appendix Section S3.1 the feasibility of using other interpretability techniques. These include gradCAM [38] and transformer raw attention [39], with which we obtain consistent results for the scene regions.

## 3 Experiments

### 3.1 Brain Data

**fMRI dataset.** We used the Natural Scenes Dataset (NSD) [5], which consists of high-resolution fMRI responses to naturalistic images from Microsoft COCO [40]. NSD contains 7T fMRI responses (1.8 mm, 1.6 s) from 8 subjects who each viewed 9,000–10,000 distinct color natural scenes (22,000–30,000 trials). Subjects fixated centrally and performed a long-term continuous image recognition task. The noise ceiling (NC) was estimated in each voxel as described in Allen et al. [5]. We only include voxels with NC $\geq 10\%$ variance. Notably, the dataset contains complex and sometimes crowded images of various everyday objects in their natural contexts at varied viewpoints. The stimulus set is thus more typical of real-world vision and allows us to characterize neural representations and computations under more ecological conditions.

**Regions of Interest (ROIs).** We used the "streams" anatomical atlas from NSD to define three ROIs that cover the parietal, lateral, and ventral visual streams (mid- and high-level ROIs were grouped). We then split each of the ROIs into sub-regions using the HCP atlas, including any ROI in the HCP atlas with at least 50% voxel overlap with any of the streams. We added a fourth ROI, which we call "Medial", and took all HCP regions overlapping with RSC. We further split the "Ventral" ROI into "Ventrolateral" and "Ventromedial", where "Ventromedial" included HCP regions overlapping with PPA. A separate network was trained for each of the HCP sub-regions. We also examined three scene ROIs – RSC, OPA, and PPA – as defined by the category functional localizer.

### 3.2 Measures

Unless otherwise specified, we evaluate on the Places365 dataset [41], and use the top 1000 images per sub-region per subject as determined by the predicted responses to the images by the sub-region network. Thus, the evaluation takes into account: 1) the top-most-activating images for a region; 2) the areas of the image that most contribute to the response as determined by network dissection. We perform brain dissection on the following measures:

**Depth.** We analyze the selectivity of depth, which refers to the perceived distance between the viewer and different regions of the scene. To obtain depth maps for each evaluation image in Places365, we use the ZoeDepth metric depth estimation network [42], which has been shown to exhibit exceptional zero-shot generalization performance on indoor and outdoor scenes. We evaluate two measures of depth: 1) *metric depth*, which represents the absolute depth in meters for each pixel in the image, and 2) *relative depth*, which denotes depth maps normalized to a range of 0-1 for each image. Consequently, absolute depth captures the overall nearness or farness across the images, while relative depth focuses on the nearness or farness within an individual image.

**Surface Normals.** Surface normals, vectors perpendicular to surfaces, are essential for spatial cognition, as they indicate surface orientation and reflect the spatial coordinate system. Using the XTC network [43], we estimate normals for each Places365 image. The x, y, and z components are segmented into 512 bins (8 per component; $8^3 = 512$ combinations), and voxel selectivity is reported as the bin number.

**3D Curvature.** Curvature features provide information about 3D structure and are invariant under rigid transformations. We use the XTC network [43] for estimating principal curvatures.

**Shading.** One way to infer scene geometry is "shape from shading" [44, 45] using the intrinsic image decomposition $I = AS$, where $S$ is a shading function parameterized by lighting and depth. We use the XTC network [43] for estimating shading.

Segment Anything [46] was used to obtain segmentation masks from the bounding boxes for each of the annotations.

# 4 Results

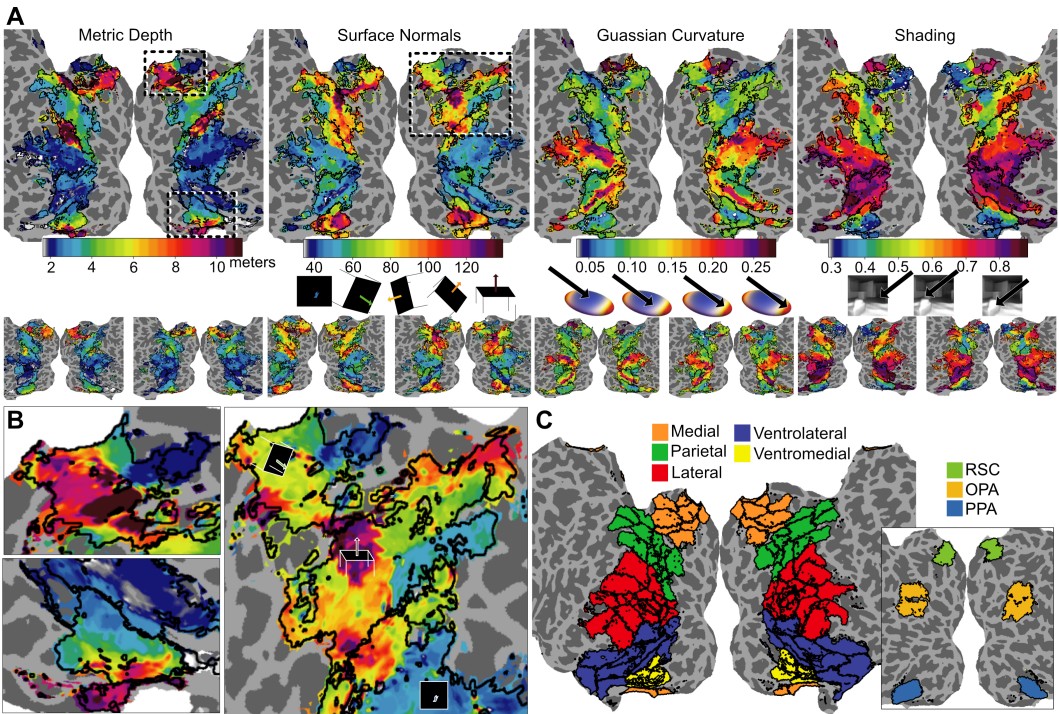

Figure 2: **A.** Depth, surface normals, Gaussian curvature, and shading on flatmaps colored by selectivity per voxel for S1 (large), S2, and S5 (maps for the remaining five subjects are in supplemental). **B.** Enlarged portions of flatmaps from A. Medial areas show depth preference for far-reaching depths (top left). There is a gradient from near to far depths from lateral to medial ventral (bottom left). Parietal and Medial areas show clusters of "upright" and "right/left" surface normals, while Lateral areas prefer "camera facing" surface normals (right).

We employed brain dissection to investigate the scene features represented throughout visual cortex. Our investigation focused on the Ventromedial, Ventrolateral, Lateral, Parietal, and Medial regions, as well as the scene network – PPA, OPA, and RSC. Our initial examination centered on identifying and quantifying the differences between these regions based on four spatial measures, namely, depth, surface normals, 3D curvature, and shading. These measures were chosen to cover a range of ecologically-relevant spatial features. Following our spatial analysis, we explored how these identified differences are reflected in the processing of object categories, attributes, and spatial relations. This aspect of our study sheds light on the relationship between the observed variations in spatial features and the resultant impact on higher-order visual processing – specifically how we perceive and interpret objects and their relationships within a given scene. Finally, we quantified the spatial clustering of voxels for each measure to better understand the spatial organization for different visual features.

## 4.1 3D Spatial Dissection

### 4.1.1 Brain Flatmaps

To gain a better understanding of how 3D structure is represented in the brain, we used brain dissection to quantify the selectivity for each voxel across the high-level visual cortex for the four

spatial measures. This process allowed us to create a detailed selectivity map for each voxel that reveals how 2D inputs are transformed into 3D representations that enable reasoning about the physical world. Figure 2 displays these selectivity maps on a cortical flatmap of the brain. We observe that Lateral areas demonstrate selectivity for close distances, aligning with the overlap of Lateral ROIs with regions selective for bodies, frequently featured in the foreground of images. Notably, we observe that the depth selectivity for the Lateral and Ventrolateral ROIs are similar, and stand in contrast with the rest of the ROIs. This pattern is also true for the other spatial metrics (Surface Normals, Gaussian Curvature and Shading) we study, where the Lateral and Ventrolateral appear as a distinct cluster from the Ventromedial, Parietal and Medial ROIs. In contrast, we observed high selectivity for far distances in the Medial and Ventromedial areas, suggesting a specialized role for behaviors that require processing and understanding far-reaching aspects of a visual scene (e.g., navigation). Additionally, we observed a strong gradient in the selectivity for depth in Ventral areas. The gradient transitioned from near-preferring depths in the Ventrolateral ROI to far-preferring depths in the Ventromedial ROI. Such a gradient is likely to be important for building up scene representations that capture a wide variety of behaviorally relevant affordances. Finally, we observed distinct patterns related to the processing of surface normals. Large areas of the Lateral and Ventrolateral regions showed a preference for surface normals in the negative X direction, pointing back at the camera (again, consistent with the preference of these areas for bodies and objects). In contrast, the Parietal and Medial areas predominantly preferred surface normals oriented in the positive Z direction (pointing upwards) and along the Y direction (pointing right and left).

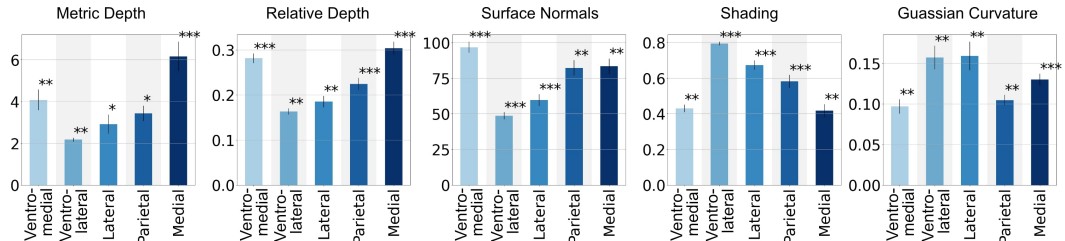

Figure 3: Mean $\pm$ standard deviation ($N = 8$ subjects) for each of the high-level visual ROIs. *** = significantly different from all other ROIs, ** = significantly different from all but one other ROI, * = significantly different from all but two other ROIs ($p < 0.01$; *post hoc* Tukey test; $N = 8$ subjects).

### 4.1.2 Spatial Dissection of High-level Visual Areas

We quantify the differences between the high-level visual ROIs in terms of mean $\pm$ standard deviation for four 3D spatial measures: depth (metric & relative), surface normals, shading, and Gaussian curvature (Fig. 3). We observe differences in depth preferences between the Medial and Ventrolateral ROIs. Medial and Ventromedial areas exhibit significantly larger absolute and relative depth selectivity compared to the Ventrolateral and Lateral regions ($p < 0.01$; *post hoc* Tukey test; $N = 8$). Conversely, Ventrolateral and Lateral areas have significantly lower average absolute depths than the Medial and Ventromedial areas, as depicted in Figure 3 ($p < 0.01$; *post hoc* Tukey test; $N = 8$). This disparity highlights the specialized roles these regions play in processing natural images across varying depth distributions, catering to both nearby and distant objects. Concerning the processing of surface normals, we observe that the Ventromedial region significantly prefers normals indicative of surfaces pointing in the "up" direction ($p < 0.01$; *post hoc* Tukey test; $N = 8$). Parietal and Medial areas show a stronger preference for surface normals oriented to the left and right compared to the other ROIs ($p < 0.01$; *post hoc* Tukey test; $N = 8$). In contrast, the Ventrolateral and Lateral regions have a preference for surface normals pointing back at the camera ($p < 0.01$; *post hoc* Tukey test; $N = 8$).

Examining the shading preferences across the ROIs, we again notice a clear distinction. Ventrolateral areas prefer the lightest shading, while Medial and Ventromedial areas prefer the darkest shading ($p < 0.01$; *post hoc* Tukey test; $N = 8$). This observation reaffirms the notion of functional segregation in the visual cortex, with different regions specializing in processing specific visual attributes, in the case of shading, perhaps reflecting additional cues to scene depth.

Lastly, we consider Gaussian curvature, a measure that captures the degree of 3D curvature of a surface. Higher Gaussian curvature implies a more spherical geometry. Ventrolateral and Lateral areas significantly prefer higher Gaussian curvature (Fig. 3), indicating a preference for more spherical

surfaces ($p < 0.01$; *post hoc* Tukey test; $N = 8$). In contrast, Parietal and Ventromedial areas prefer lower Gaussian curvature, suggesting a preference for flatter surfaces ($p < 0.01$; *post hoc* Tukey test; $N = 8$). Ventrolateral and Lateral areas may be encoding spatial features relevant to object appearance for both recognition and grasping, while Parietal and Ventromedial areas may be encoding scene layout in terms of large-scale surfaces and scene elements. Such findings add another layer to our understanding of how different areas of the visual cortex process and interpret 3D scene structure.

### 4.1.3   Spatial Dissection of Scene Areas

In this section, we delve into the differences between the scene ROIs (PPA, OPA, RSC) with respect to the four 3D spatial measures, providing a quantified analysis of their specialized roles in scene perception. We illustrate the voxel-selectivity on brain flatmaps (Fig. 4A) and plot the mean $\pm$ standard deviation ($N = 8$) for each of the each ROIs and measures (Fig. 4B). These distributions and mappings elucidate the distinct patterns of selectivity across the scene ROIs.

There is a considerable increase in both metric and relative depth for RSC compared to both PPA and OPA, as evidenced in Figure 4B ($p < 0.001$; *post hoc* Tukey test; $N = 8$). This pattern indicates that RSC in processing far-reaching information in scenes, consistent with the claim that RSC is preferentially involved in navigation [47]. Additionally, we observe a distinct shift in surface normal preference, with RSC showing a preference for more right/left surface normal bins compared to OPA and PPA, which favor upright surface normals ($p < 0.001$; *post hoc* Tukey test; $N = 8$). This distinction underscores the functional segregation between these scene ROIs in interpreting surface orientation and suggest, again, that RSC may represent information supporting navigation, while OPA and PPA may represent information more relevant to interacting within a local region of space.

To illustrate these observations, we identified voxels that have the closest selectivity to the ROI mean for depth or surface normals for PPA, OPA, and RSC, and plotted the voxel's top five images by predicted response (Fig. 5). We include corresponding dissection masks below these on images and either depth or surface normal images. RSC exhibits a preference for far depths and more "right/left" surface normals, indicative of vertical structures relevant to navigation. In contrast, OPA and PPA exhibit a preference for "up" surface normals which are relevant to representing the local scene layout. Once more, these results highlight the distinct functional roles these areas play in scene perception.

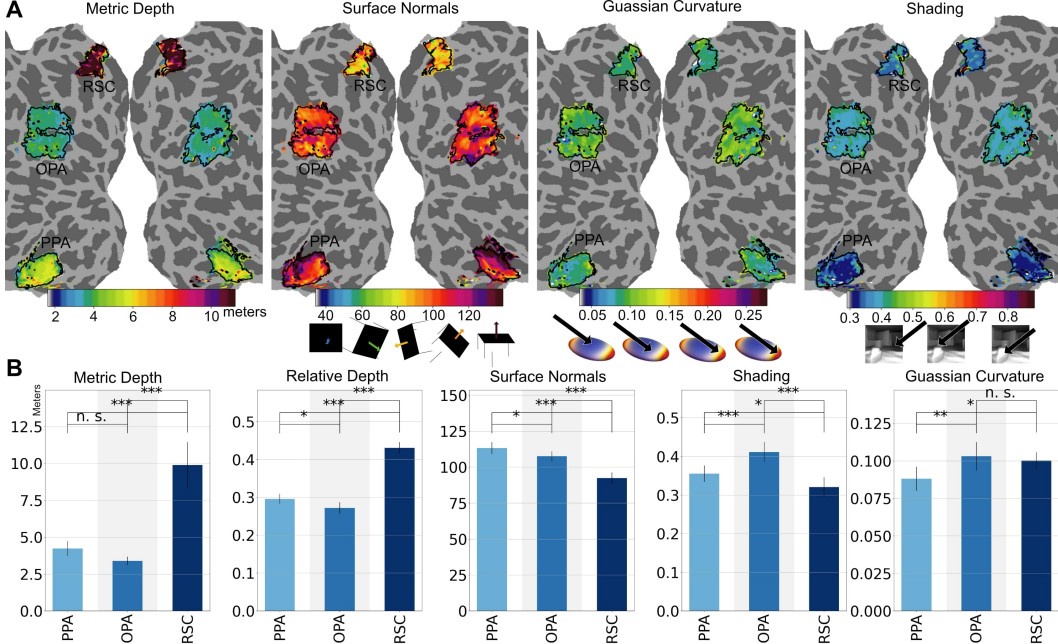

Figure 4: **A.** Flatmaps showing voxel-preferences in OPA, PPA, and RSC for the four spatial measures. **B.** Mean $\pm$ standard deviation ($N = 8$ subjects) for each of the high-level visual ROIs. *** $= p < 0.001$, ** $= p < 0.01$, * $= p < 0.05$, n.s. = not significant $p > 0.05$ (*post hoc* Tukey test; $N = 8$ subjects).

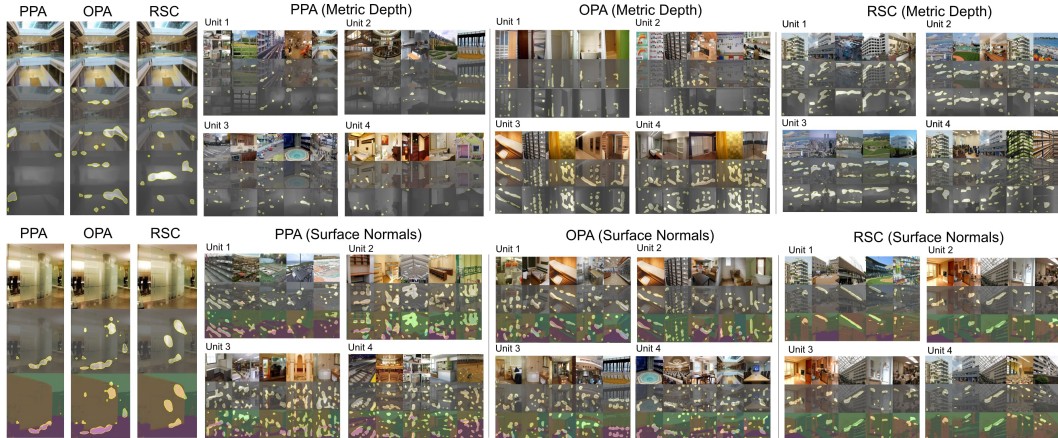

Figure 5: Voxels that have the closest selectivity (by euclidean distance) to the ROI mean of depth (top) or surface normals (bottom) for PPA, OPA, and RSC. An example image is shown with a representative voxel (left) and the top five images by predicted response for the top four voxels (right) for each ROI. Dissection masks are visualized on the images and either the depth or surface normal images. RSC prefers far-reaching depth and vertical structures, while OPA and PPA prefers more local depths and a mix of vertical and horizontal structures.

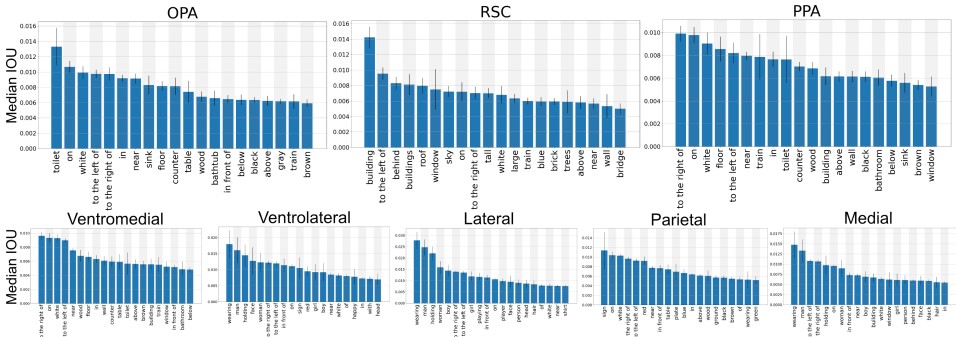

Figure 6: Median IOU (mean $\pm$ standard deviation; $N = 8$ subjects) for the GQA dataset [48] for each of the scene ROIs (top) and high-level visual ROIs (bottom).

## 4.2 Object Relations, Attributes and Categories

We utilize the GQA dataset [48] in concert with brain dissection to gain insights into the selectivity differences for object relationship, attributes, and categories within an image. The median IOU (mean $\pm$ standard deviation across 8 subjects) for the top 20 concepts for each ROI is reported in Figure 6. With respect to high-level visual ROIs, we observe distinct preferences: Parietal areas prefer inanimate and scene objects, such as walls and tables, that are more relevant to both navigation and physically interacting with objects. In contrast, Lateral areas exhibit a preference for person and animate objects, while Ventrolateral areas favor persons and objects – representations that support social interactions and scene understanding. When examining object relations, Parietal areas showed a strong preference for spatial relations such as 'in front of,' 'near,' 'in,' and 'above'. Lateral areas favor relations involving people, such as 'wearing,' 'holding,' and 'with'. In terms of attributes, Parietal areas show a preference for colors and materials, whereas Lateral areas favor animate attributes involving people, such as 'playing,' 'standing,' and 'happy'. These preferences are largely aligned with prior results showing Parietal preferentially processes spatial visual information [22], Lateral preferentially processes social visual information [24], and Ventral preferentially processes objects and faces [23].

As illustrated in Figure 6, within scene ROIs, OPA shows greater selectivity for indoor scene elements, such as chairs, tables, and floors. PPA exhibits a mix of indoor and outdoor preferences, whereas RSC shows a clear preference for outdoor scene elements like trees, buildings, and the sky, once again reinforcing a dichotomy between local space interactions and navigation. Furthermore, when

considering attributes, RSC again demonstrates more selectivity for "outdoor" attributes compared to PPA and OPA, which show higher selectivity for indoor elements. These findings underscore the distinct functional implications of these ROI selectivity profiles for processing visual scenes.

## 4.3 Topographic Organization of the Spatial Preferences

We investigated the topographical organization of the spatial measures within each ROI. High-level visual areas are topographically organized, comprising multiple hierarchically organized areas that are selective for specific domains such as faces and scenes [49, 50, 12, 13, 14, 15, 16, 51]. However, whether and how this topographical organization applies to the spatial measures used here remains unclear. Given a specific type of spatial information, we explored: (a) if the voxels in an ROI are differently tuned for this type of information (e.g., sub-groups of voxels are tuned for different surface normal orientations); and if so, (b) does the tuning follow a smooth topographical organization with relatively large clusters processing similar information. To quantify the degree of clustering by selectivity similarity, for each measure we computed the average cluster proportion (cluster size normalized by total voxels in the ROI) (considering clusters > 5 voxels) that each voxel belongs to, and report the results compared to a random permutation of the voxels (Fig. 7). We employed connected components [52] on the 3D voxel data to determine whether two neighboring clusters belong to the same cluster. For continuous measures, such as depth, we consider two voxels the same if they fall within $\delta$ of each other. We used $1/100^{th}$ of the data range across all ROIs as the $\delta$ for each of the continuous measures. For discrete concepts, such as object category, we consider two neighbors to be in the same cluster if they belong to the same discrete category.

A cluster analysis was run for each ROI. We report the average cluster size for a given measure compared to the average cluster size for the measure when the voxels are randomly permuted ((Fig. 7; $p < 0.05$ paired $t$-test; $N = 8$)). We visualize the spatial clusters on flatmaps: All of the high level visual areas showed clustering significantly different from random for all spatial measure, indicating that their voxels are tuned for different values of each measure in a way that is topographically organized. At the level of the scene areas however, RSC was not distinguishable from chance for metric depth, relative depth, shading, gaussian curvature and category. This suggests that except for surface normals, the voxels in RSC are tuned similarly. Similarly, the results suggest that the voxels in PPA are tuned differently only for metric depth, and the voxels in OPA are tuned differently for metric depth, relative depth and surface normals. These findings highlight the robust topographical organization within the ROIs, extending beyond traditional domain-specific areas. Critically, this extended organization highlights the fact that, as a whole, visual cortex is representing a broad range of ecologically relevant properties that support complex behavior [53].

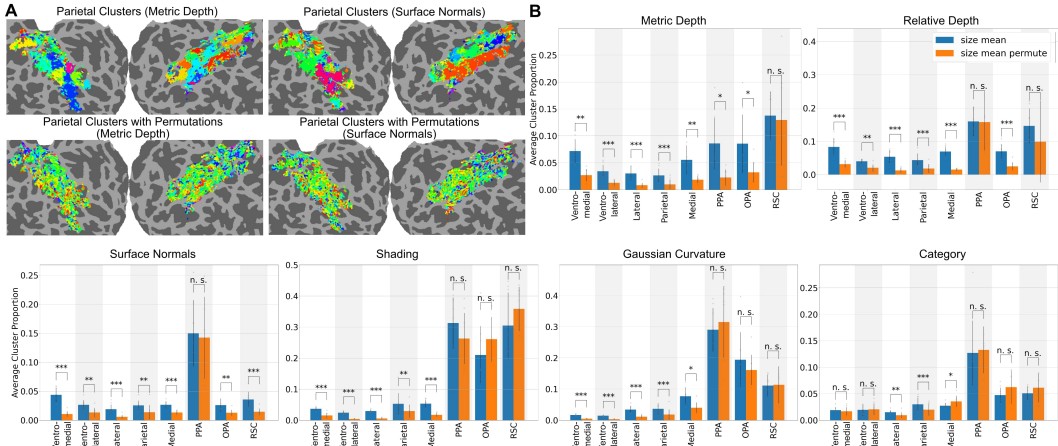

Figure 7: **A.** Metric depth (left) and surface normals (right) spatial clusters on flatmaps for example subject S1, where each color is a different cluster with (bottom) and without (top) random spatial permutations. **B.** Bar plots showing mean ± standard deviation of average cluster size per voxel for each ROI. *** = $p < 0.001$, ** = $p < 0.01$, * = $p < 0.05$, n.s. = not significant $p > 0.05$ (paired $t$-test; $N = 8$).

# 5 Conclusion - Discussion

A clearer understanding of representational progression in natural vision requires developing hypothesis-neutral models and interpretable methods. We leverage advances in quantifying feature selectivity of artificial neural networks [6], merged with response-optimized models trained on fMRI data [7], to better characterize spatial properties of a broad range of visual brain regions. This method, "brain dissection," enables granular examination of feature preferences within multiple visual processing streams and functional sub-regions like the scene network. Contrary to earlier studies [7], our work focuses on characterizing visual sub-region preferences in terms of spatial properties and high-level categories (e.g., faces, places). Dissecting spatial properties provides complementary perspectives on visual processing: relationships, orientations, positions of objects, and surfaces within a scene in terms of spatial features (see also [10]).

We observed specific preferences in regions like the RSC, OPA, and PPA. The RSC demonstrated a pronounced preference for greater depths, outdoor object categories, attributes, relations, and predominantly "right/left" surface normals. In contrast, the OPA exhibited preference for proximate depths, intricate 3D geometries, indoor scene object categories, relations, attributes, and a higher inclination towards "upward" surface normals. These findings align with literature suggesting OPA is primed for local navigational affordances [20, 34], while RSC is geared towards facilitating landmark-based spatial-memory retrieval [54, 55]. The PPA preferences spanned a middle ground between OPA and RSC, slightly leaning towards OPA, supporting its role in encoding scene structure at a coarser scale than OPA [34]. A salient finding is RSC's pronounced selectivity for vertical surface normals, contrasting with PPA and OPA's preference for horizontal supporting structures, like tabletops and floors, highlighting PPA's sensitivity to scene layout and surface arrangement [56].

Further distinguishing between high-level visual ROIs, our study revealed gradients from ventro-lateral to medial areas. Ventro-lateral regions exhibited a preference for closer depths, predominantly horizontal surface normals, and darker shading, while medial areas showed opposite preferences, aligning with distinct processing of foreground objects and distant background elements in these pathways [54]. Added granularity was evident in variability in depth and surface normal selectivity across voxels in medial and parietal regions, indicative of specialized regions for different 3D profiles and global shape processing [57, 58]. This pattern may also reflect differences in coordinate systems for spatial information representation. Ventral areas' preference for camera-oriented surface normals indicates an egocentric frame of reference, while medial and parietal areas' preference for ground plane-oriented surface normals indicates an allocentric frame of reference. Parietal areas favored inanimate objects and spatial relations, Lateral areas preferred persons and animate objects, and Ventral areas favored both, aligning with previous literature. Furthermore, the parietal region's pronounced selectivity for spatial relations (like 'on', 'near', etc.) underscores the significance of spatial relation encoding for this area, corroborating recent research [58].

**Limitations.** Our approach relies on network dissection and therefore inherits some of its limitations. Network dissection considers each output unit in the network (in our case corresponding to an fMRI voxel) in isolation during interpretation, thus it is not capable of identifying the selectivity of a group of output units. We thus identify the selectivity of individual voxels, which is in line with the voxel-wise modeling school of thought, but not with multivariate pattern analysis. Furthermore, in natural images, there are inherent correlations between specific categories and their spatial attributes (e.g., scenes tend to be rectilinear, pointing up, with far away features, while bodies tend to be close up, facing the camera, etc. [10, 11]). Lastly, our approach relies on pretrained networks to derive spatial measures from images, and this can introduce some estimation errors. We mitigate this limitation by focusing on ROIs that are widely held to encode spatial properties.

A comprehensive understanding of natural vision should cover early visual processing, ecologically-based intermediate features, and high-level, semantically-grounded representations. Current knowledge largely focuses on the first and third areas. Brain dissection offers potential for uncovering data-driven insights into the full visual processing hierarchy.

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

## S1 Overview

Section S2 contains more details of the methods described in the main paper. Section S3 provides additional results from the experiments.

We include an interactive project page, which includes exploratory brain flatmaps for each of our dissection measures, and code for training and evaluating the networks. Please visit the following link to access the project page: https://brain-dissection.github.io/

## S2 Implementation details

### S2.1 Network Architecture and Training

Our architecture and training largely follow Khosla et al. [59], except we use regular convolutions instead of E(2)-steerable convolutions. We found that using regular convolutions instead of E(2)-steerable convolutions does not significantly affect dissection performance while providing significantly improved training speed and memory efficiency. We train a separate convolutional neural network (CNN) for each ROI sub-region shared across all 8 subjects. The CNN consists of four "blocks", each with two convolutional layers followed by average pooling. We use batch normalization and early stopping. For each voxel in the ROI sub-region (across for all subjects), we employ a linear readout model on top of the feature space to predict the responses of individual voxels in a specific brain region. The linear readout is factorized into spatial and feature dimensions following [37]. This allows us to separate image spatial tuning (the "where") from feature tuning (the "what"). The spatial features have been shown to correlate with the population receptive fields (pRFs) of the voxels [21]. We train with a mean squared error loss between the predicted voxel response and the true voxel response. During the dissection procedure, we only use the feature tuning. We train all models on a single Nvidia GeForce RTX 3090 GPU. The number of parameters in the CNN is 784697 and the number of parameters in the transformer network (see next section) is 7053009.

### S2.2 Training & Validation Curves

We provide encoding model training and validation curves for PPA, OPA and RSC functionally-defined ROIs in Figure S1 and Figure S2. We utilize early stopping to terminate training and obtain an evaluation checkpoint by checking if the validation performance (Pearson correlation) is lower than the highest validation performance for 10 epochs.

### S2.3 Additional Evaluation Details

For network dissection, we use the code repository from Bau et al. [6]. We use pycortex to visualize voxel data [60]. Data are projected onto 2D brain flatmaps from 3D voxel data using trilinear interpolation for continuous data (e.g., depth) and nearest interpolation for discrete data (e.g., category).

## S3 Additional Results

### S3.1 Performance of Response-Optimized and Baseline Models

We have provided the Pearson correlation coefficient plotted on a flatmap for all ROIs on a held-out test set of NSD data in Figure S3 for example Subject S1. Additionally, we compare our model's mean Pearson correlation to the features from Lescroart and Gallant [20] and ImageNet task-optimized features [61] on held-out NSD brain data using Pearson correlation. We the features fit to brain data via Ridge regression. We computed the features in Lescroart and Gallant [20] for NSD images using the estimation networks. As shown in Figure S3, our brain dissection model demonstrates a significant improvement, nearly 2x across scene ROIs, in its alignment with the brain responses compared to the baseline features from [20]. Our model also aligns in performance with AlexNet ImageNet features, even though the AlexNet network having a significantly larger parameter size and trained on significantly more images.

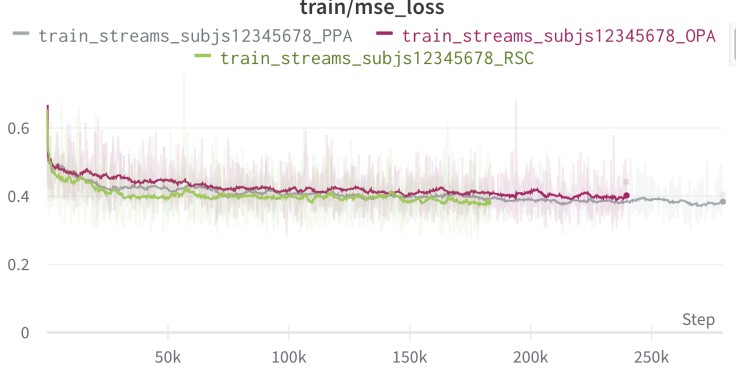

Figure S1: Mean squared error training curves for encoding models for PPA (gray), OPA (pink), and RSC (green) ROIs. X-axis is number of training steps.

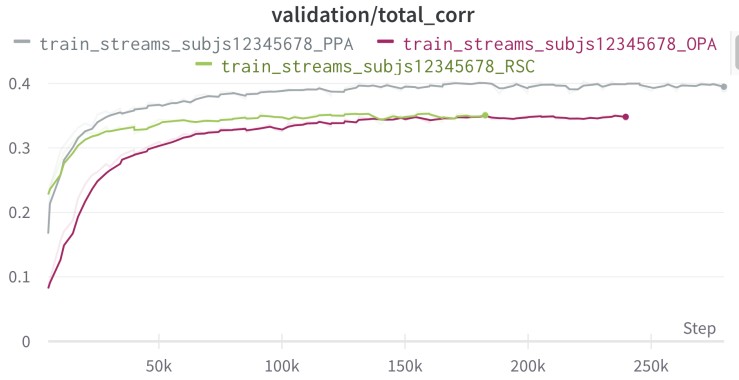

Figure S2: Pearson Correlation validation curves for encoding models for PPA (gray), OPA (pink), and RSC (green) ROIs. X-axis is number of training steps. Training was terminated using early stopping based on the validation curves.

## S3.2 Results using Different Architectures and Interpretability Methods

We integrated both gradCAM [38] and raw attention techniques [39]. The former utilizes input features combined with network gradients, while the latter leverages attention scores from transformer architectures. As shown in Figure S4, our results affirm consistency across diverse interpretability methods and network architectures.

## S3.3 GQA Data Visualization

We show example images and annotations for the GQA dataset in Figure S5. We used Segment Anything [46] to obtain the segmentation masks from the GQA box annotations.

## S3.4 Histograms of the Spatial Measures

We show the distribution of voxel preferences for each high-level visual ROI in Figure S6 and for the scene ROIs in Figure S7. We observe that subjects are largely consistent in their distributions for each ROI. We also observe that some ROIs and measures are multi-modal, such as in several measures in Medial and Gaussian curvature in Ventro-lateral and Lateral. This may indicate a further specialization of voxel clusters in these areas.

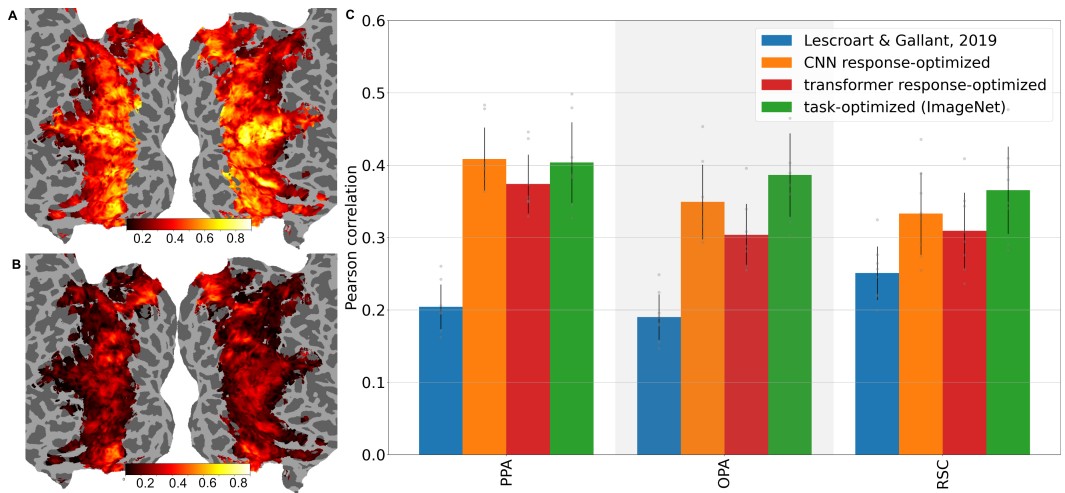

Figure S3: **A.** Pearson correlation coefficient plotted on a flatmap for all ROIs on a held-out test set of NSD data for example Subject S1 for our response-optimized networks. **B.** Same as **A** but for the features of Lescroart and Gallant [20]. **C.** Mean Pearson correlation for our model, the features from Lescroart and Gallant [20], and the ImageNet task-optimized features [61] on held-out NSD brain data using Pearson correlation for the scene-selective ROIs.

### S3.5 Places365 Evaluation

In addition to the GQA dataset, we evaluate for category selectivity in the Places365 dataset, where segmentation masks are obtained using the Unified Perceptual Parsing image segmentation network [62] previously trained on 20,000 scene-centric images from the ADE20k dataset [63]. We report median IOU (mean ± standard deviation across 8 subjects) in Figure S8.

### S3.6 Unit Visualizations

We include additional single-voxel visualizations for RSC, OPA, and PPA in Figure S9, Figure S10, and Figure S11, respectively.

### S3.7 Brain Flatmaps

In this section, we include brain flatmaps for all eight subjects for metric depth (Figure S12), relative depth (Figure S13), surface normals (Figure S14), Guassian curvature (Figure S15), and shading (Figure S16).

### S3.8 WordCloud Visualizations

We offer a WordCloud representation to visualize the top 20 categories that an ROI selects for. In this visualization, the category size represents the magnitude of the median IOU. We show WordClouds for all high-level visual ROIs and scene ROIs for the GQA dataset (Figure S17) and the Places365 dataset (Figure S18).

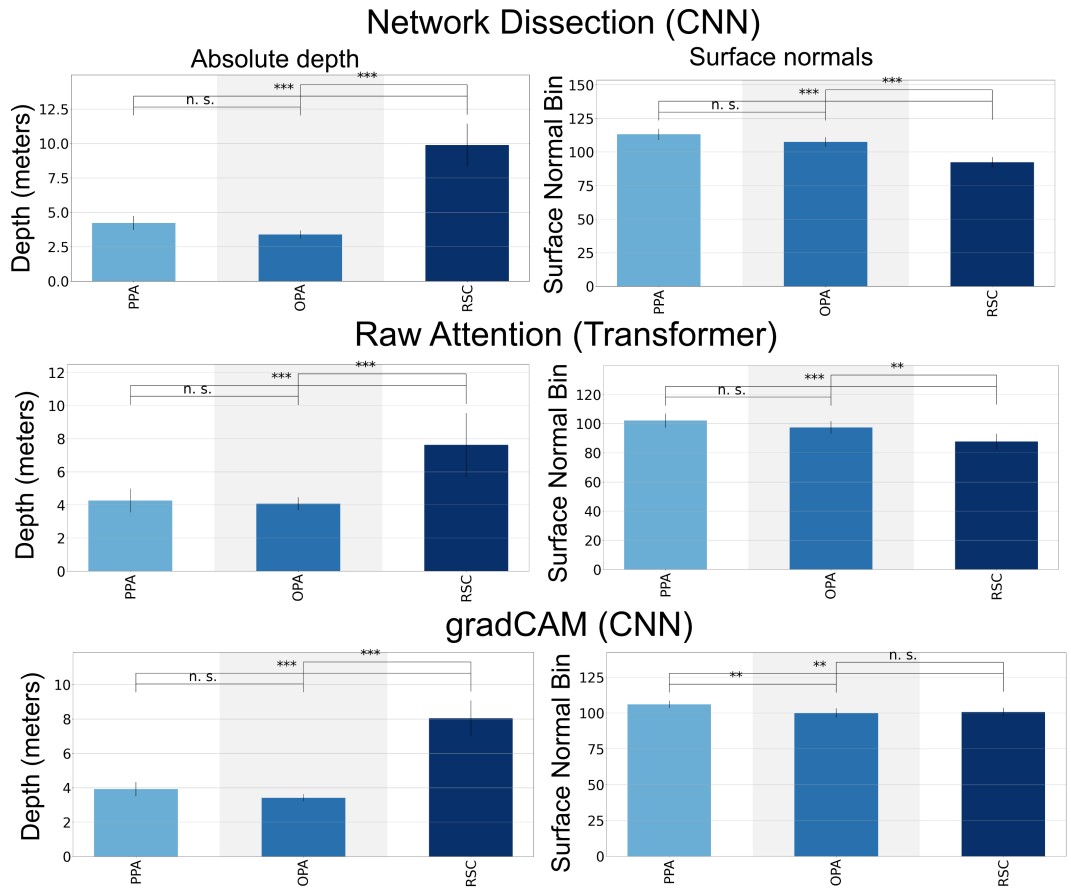

Figure S4: Depth and surface normal results for the Network Dissection [6], gradCAM [38] and raw attention techniques [39] for the scene-selective ROIs.

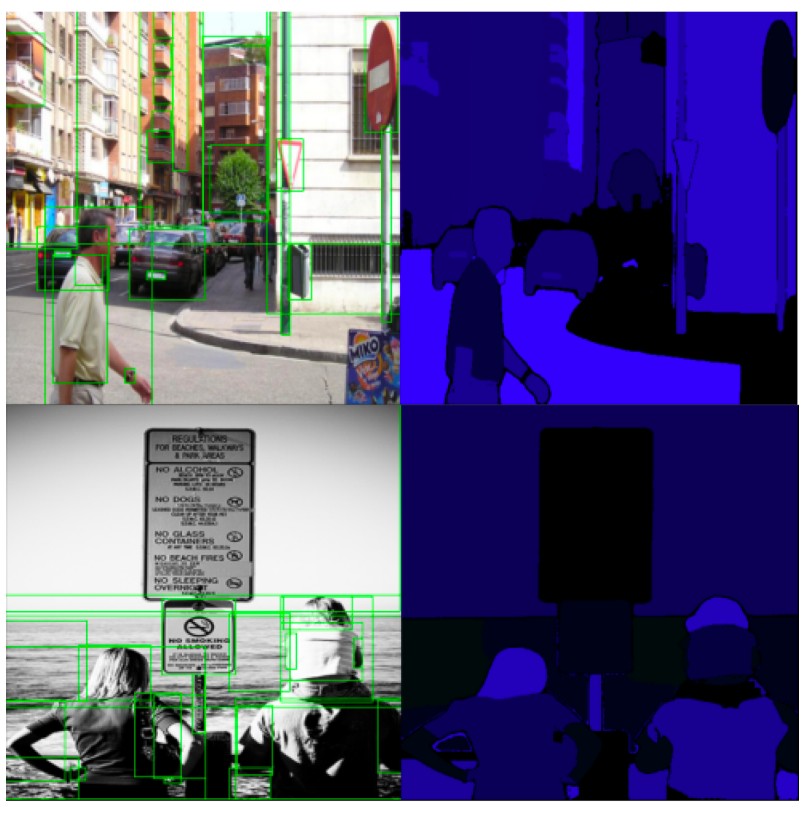

Figure S5: GQA image with bounding box annotations (left). Segment Anything [46] output given the box annotations as input (right).

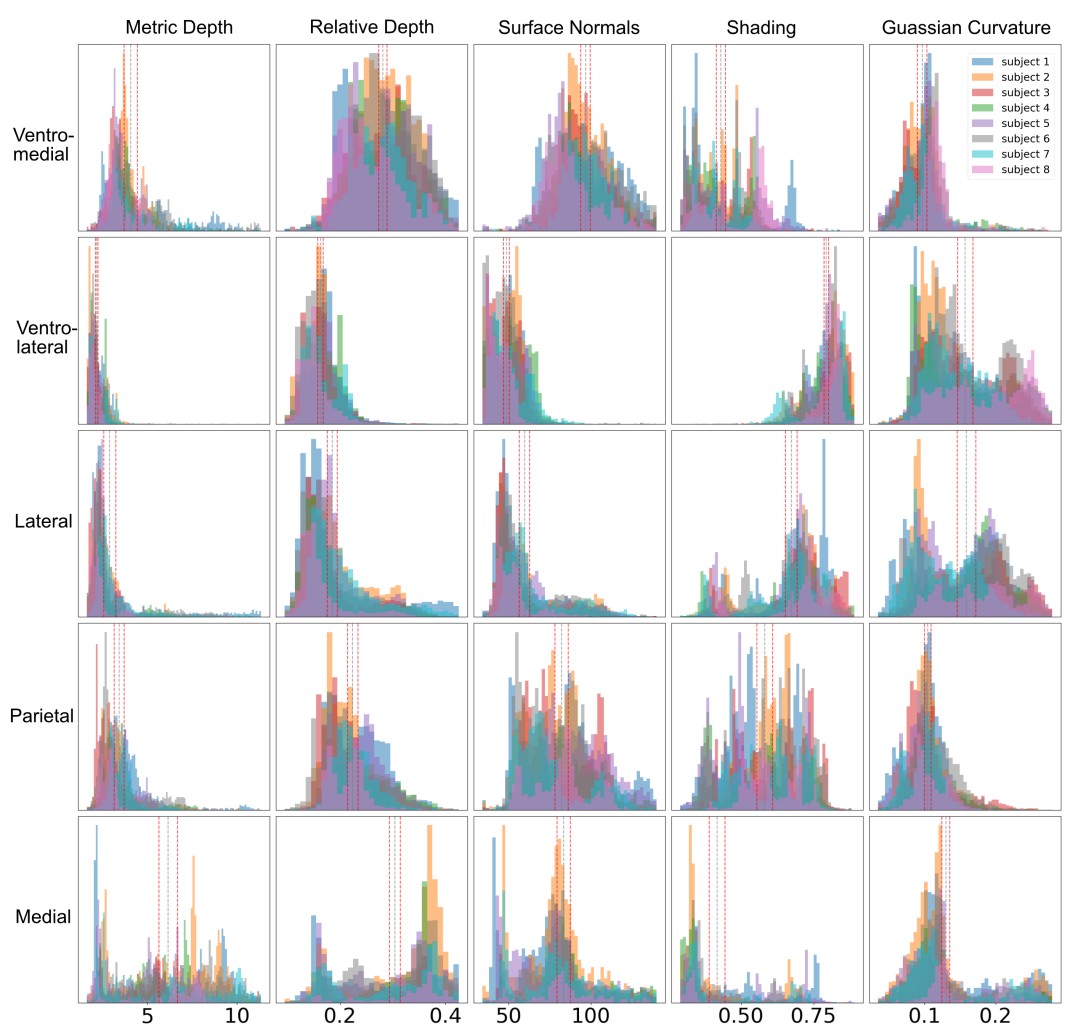

Figure S6: Histogram representation of each measure for each high-level visual ROI. Each color represents a different subject. Gray vertical dotted lines indicate the grand mean, and red vertical dotted lines indicate the 95% confidence intervals.

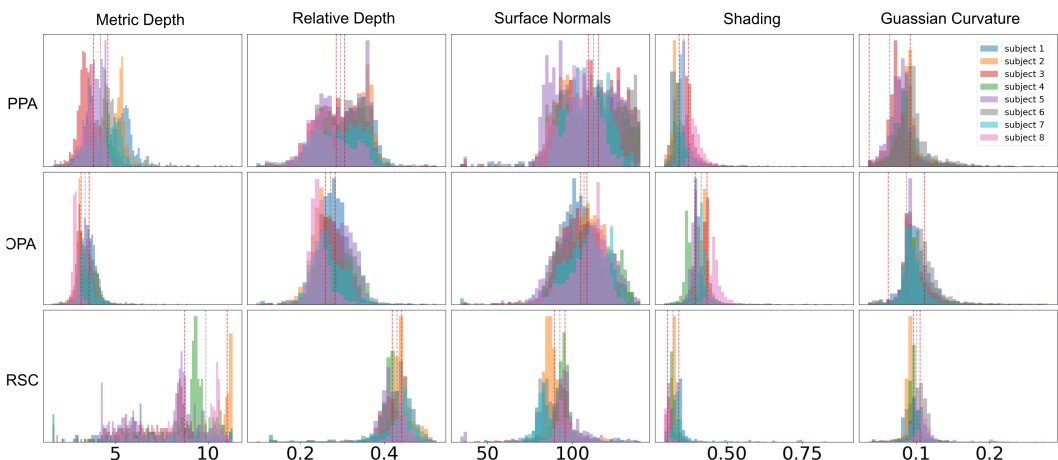

Figure S7: Histogram representation of each measure for each scene ROI. Each color represents a different subject. Gray vertical dotted lines indicate the grand mean, and red vertical dotted lines indicate the 95% confidence intervals.

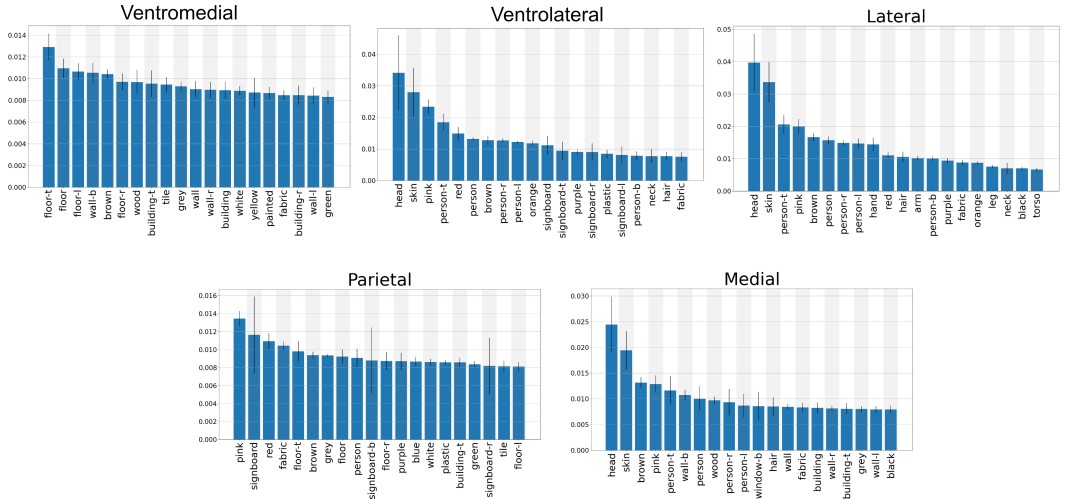

Figure S8: The median IOU (mean ± standard deviation across 8 subjects) for the top 20 concepts for each high-level visual ROI for the places365 dataset.

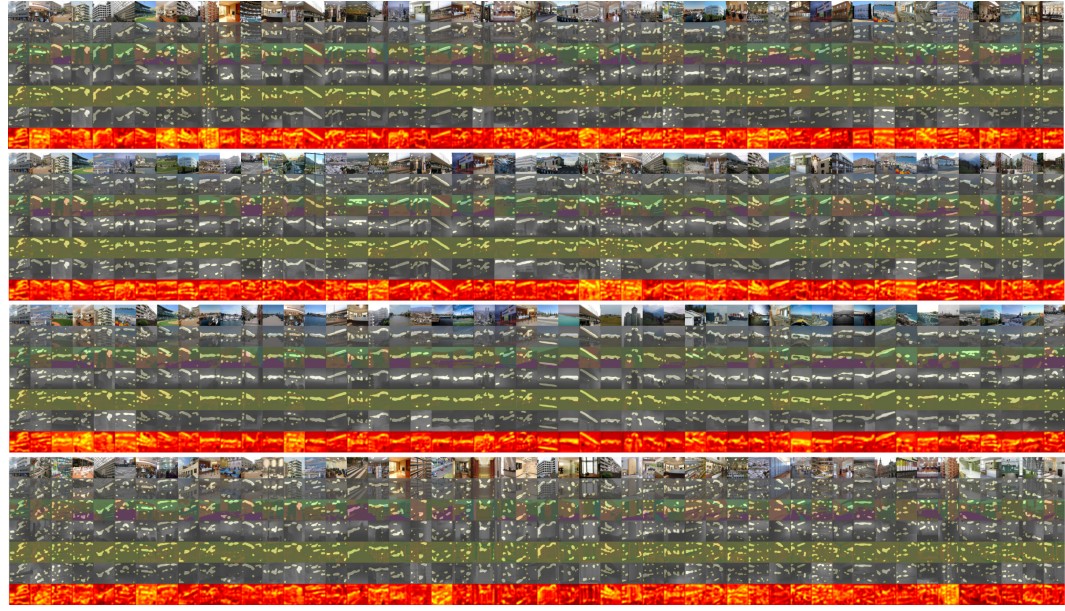

Figure S9: Unit visualization plots for four additional RSC units aligned with the mean depth selectivity of the ROI (note images are down-sampled to save memory). Each unit displays its top 50 images by predicted response for that unit. For the 50 images, we show 7 sets of images (by row - top to bottom): 1. original RGB; 2. Dissection mask overlaid on the RGB image; 3. Dissection mask overlaid on the surface normal map; 4. Dissection mask overlaid on the depth map; 5. Dissection mask overlaid on the Gaussian curvature map; 6. Dissection mask overlaid on the shading map; 7. heatmap visualization of the voxel feature map for the image.

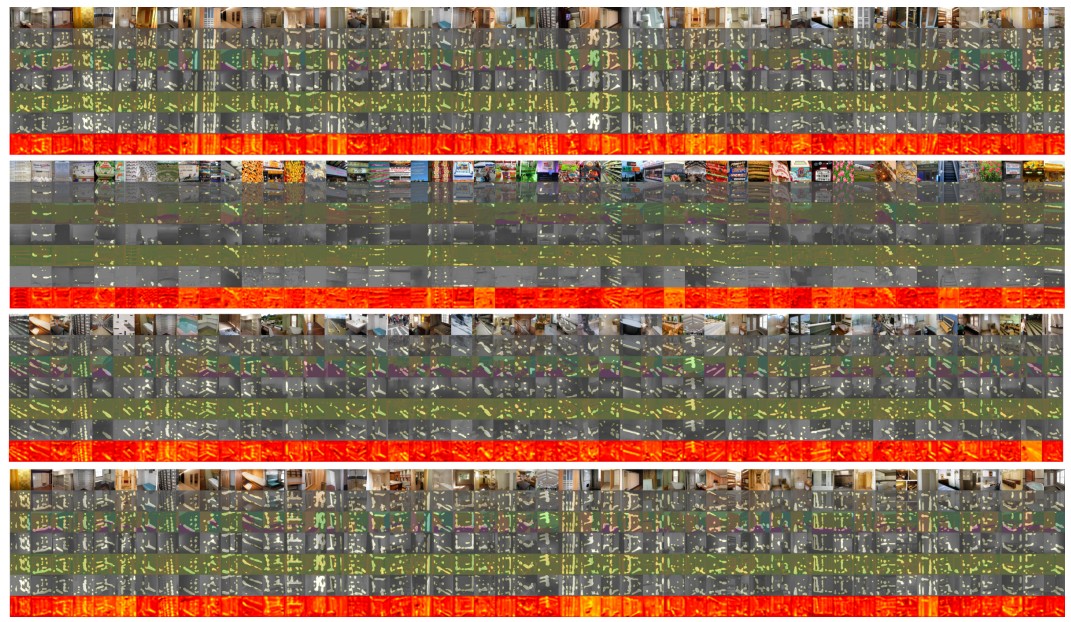

Figure S10: Unit visualization plots for four additional OPA units aligned with the mean depth selectivity of the ROI (note that images are down-sampled to save memory). Same format as Figure S9.

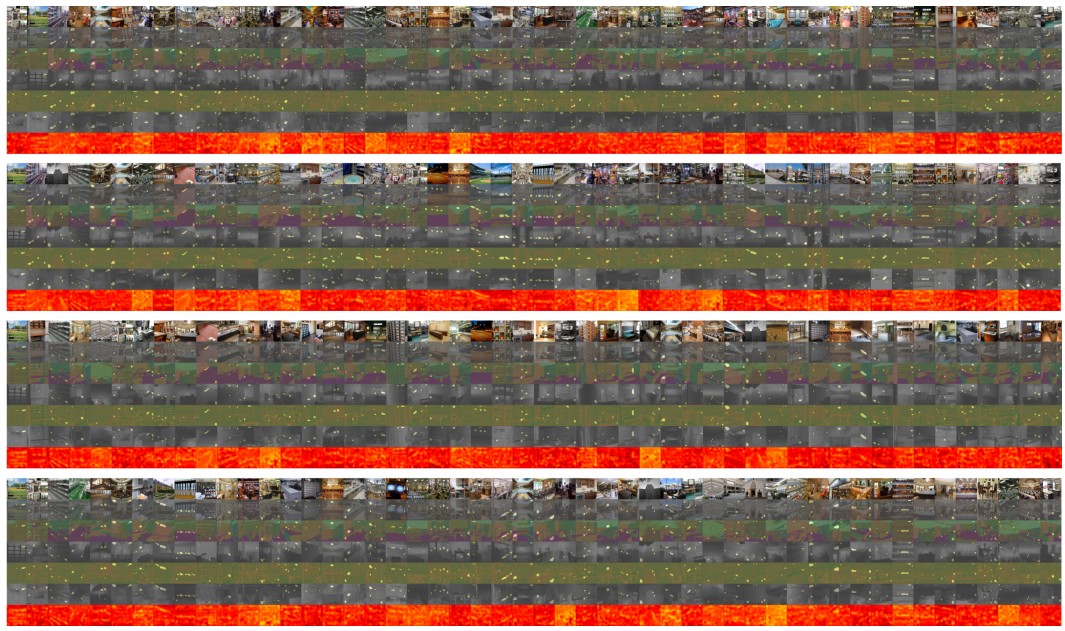

Figure S11: Unit visualization plots for four additional PPA units aligned with the mean depth selectivity of the ROI (note that images are down-sampled to save memory). Same format as Figure S9.

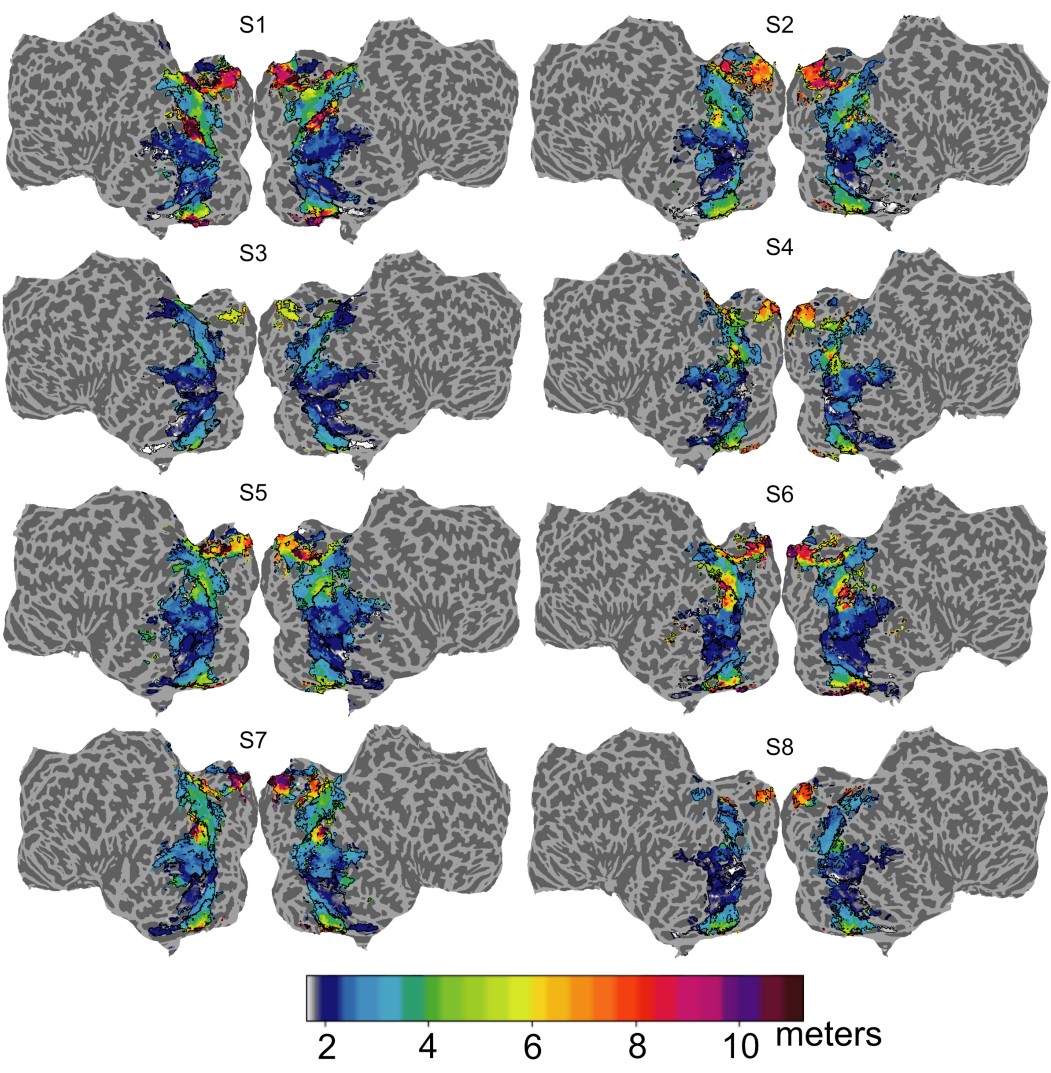

Figure S12: Metric depth on flatmaps colored by selectivity per voxel for subjects S1-S8.

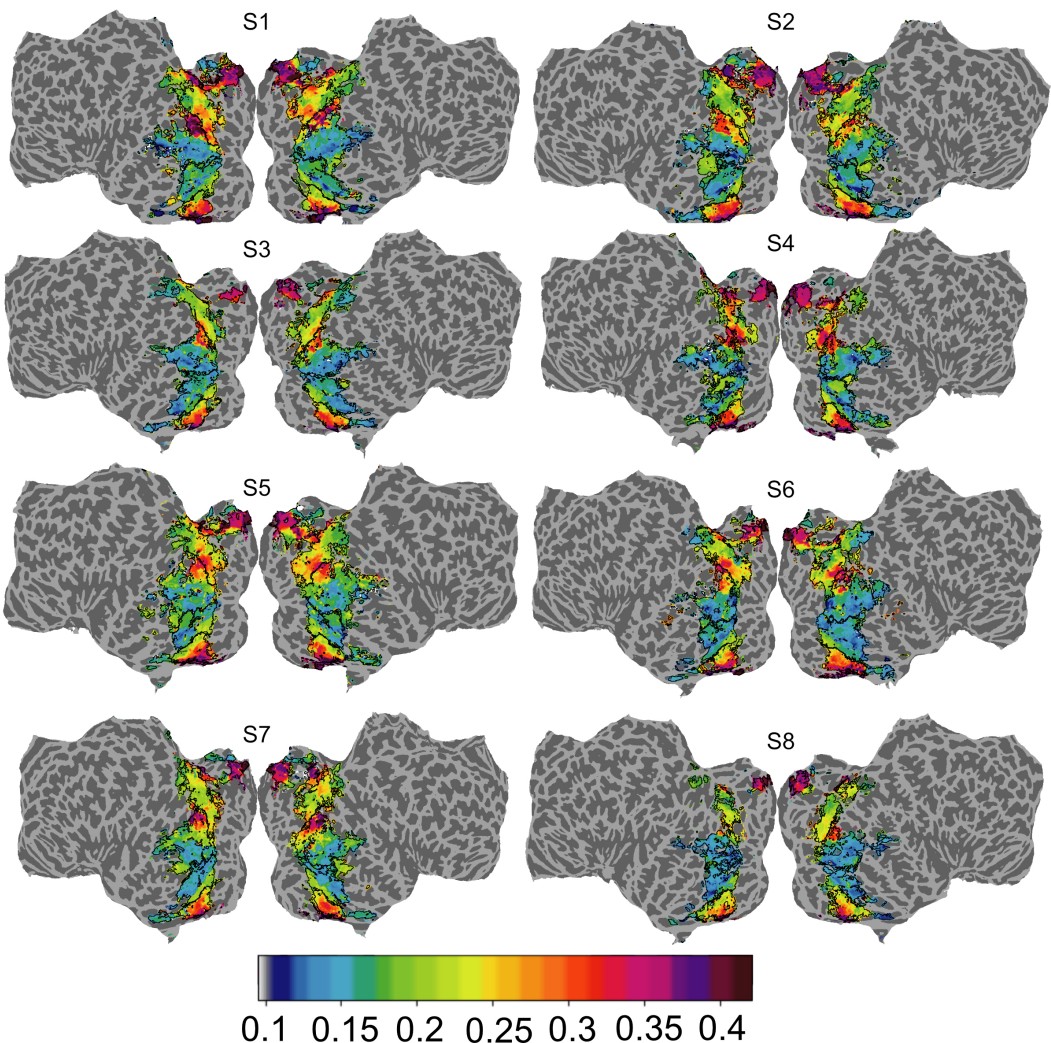

Figure S13: Relative depth on flatmaps colored by selectivity per voxel for S1-S8.

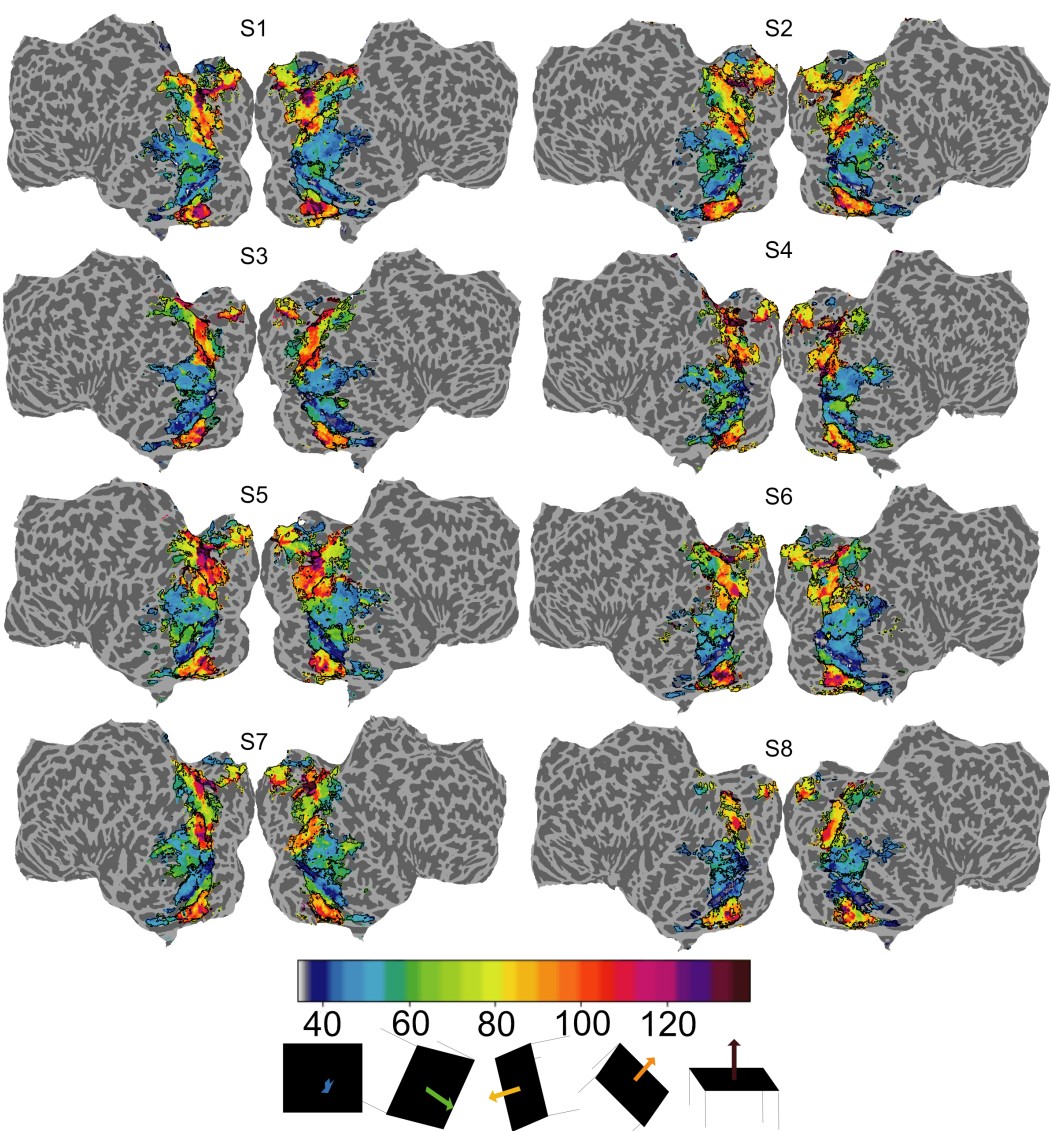

Figure S14: Surface normals on flatmaps colored by selectivity per voxel for S1-S8.

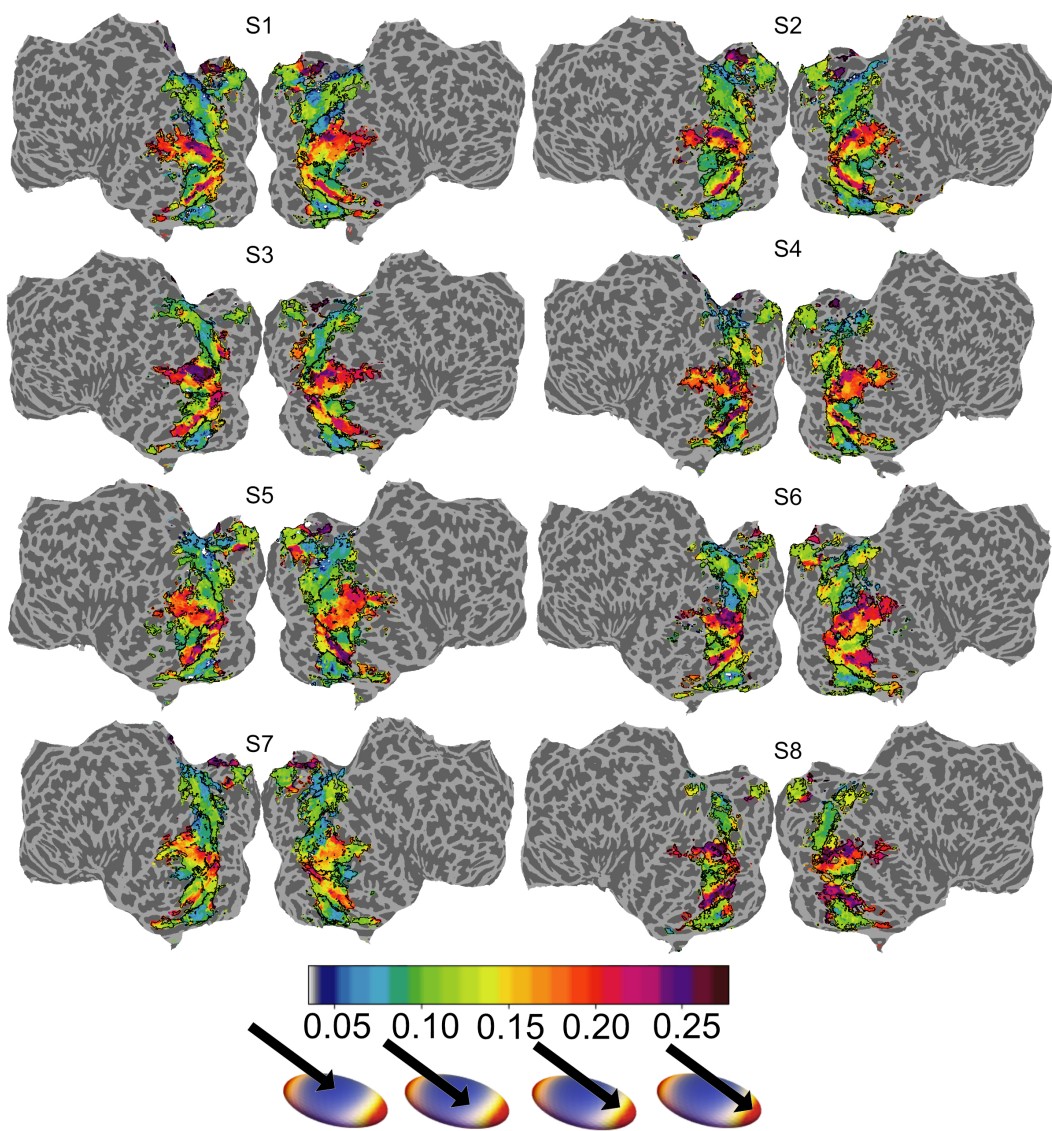

Figure S15: Guassian curvature on flatmaps colored by selectivity per voxel for S1-S8.

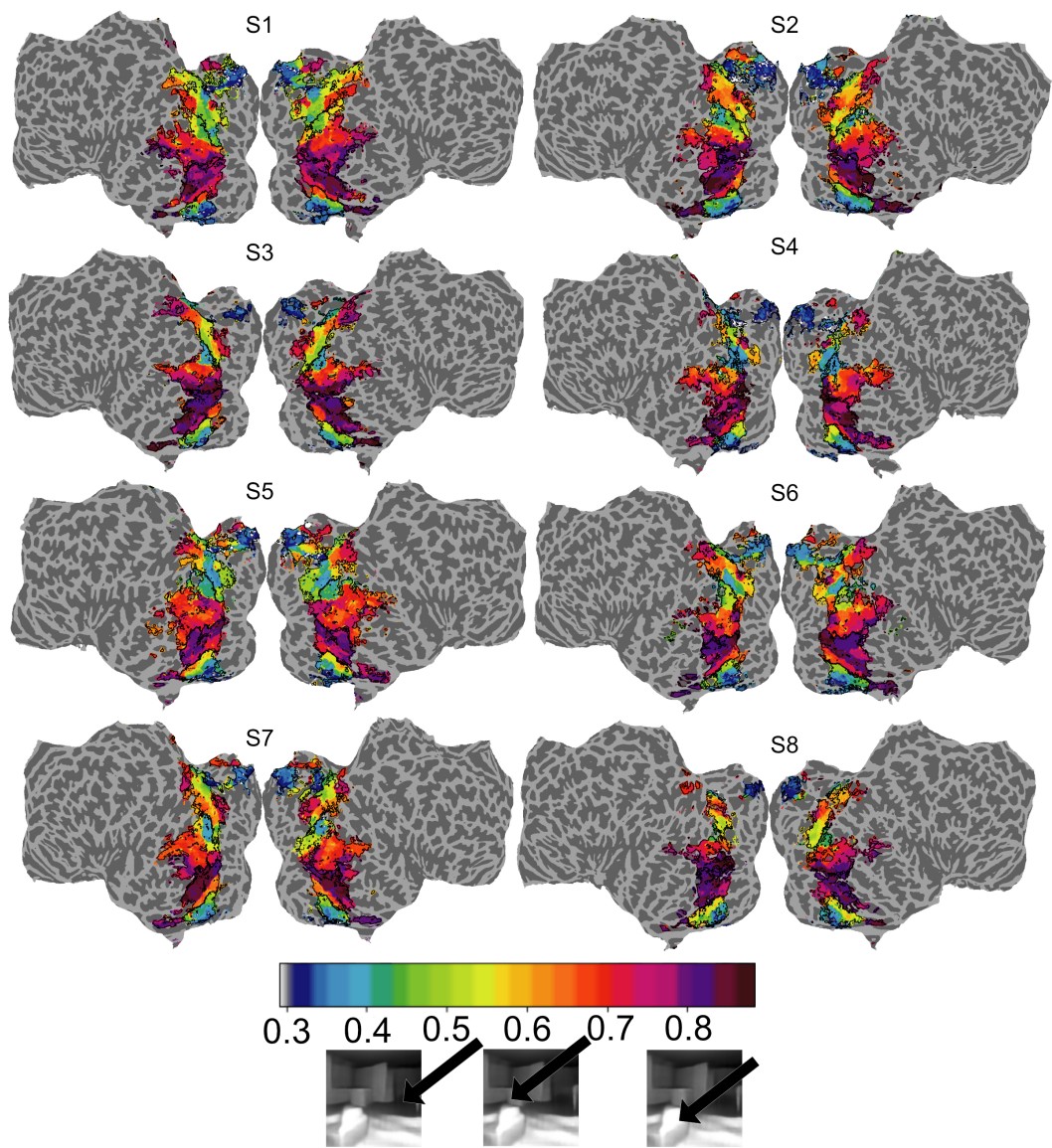

Figure S16: Shading on flatmaps colored by selectivity per voxel for S1-S8.

## Ventrolateral

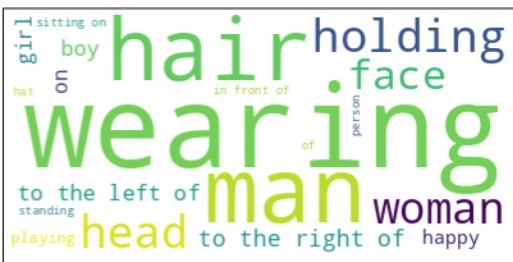

## Ventromedial

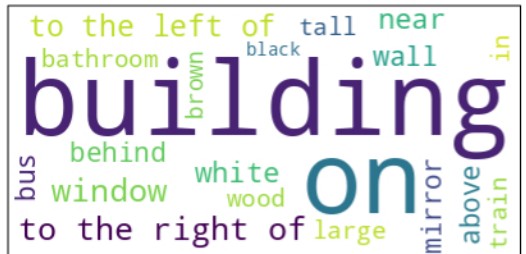

## Lateral

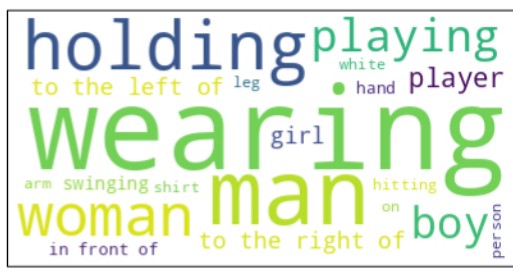

## Parietal

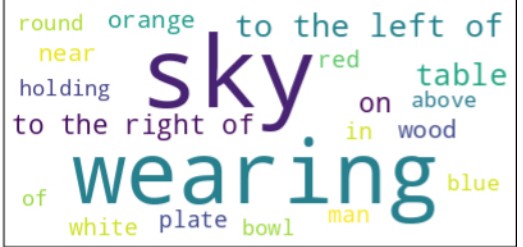

## Medial

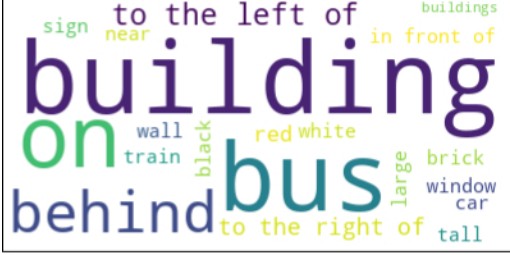

## RSC

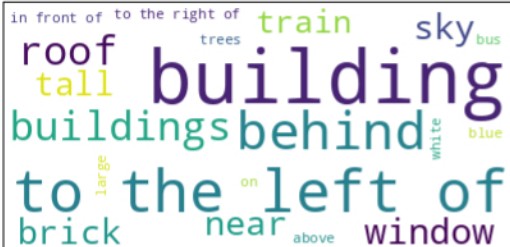

## OPA

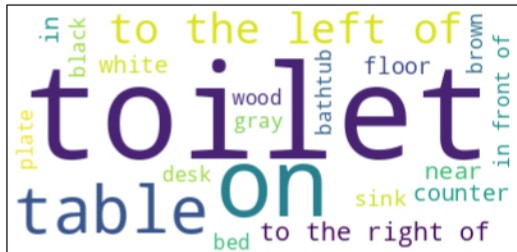

## PPA

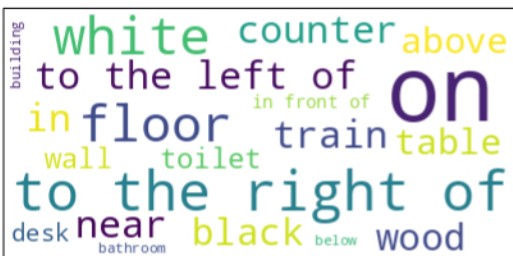

Figure S17: WordCloud for top 20 categories that an ROI selects for in the GQA dataset. In this visualization, the category size represents the magnitude of the median IOU

## Ventrolateral

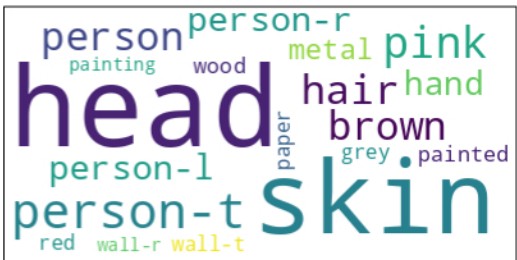

## Ventromedial

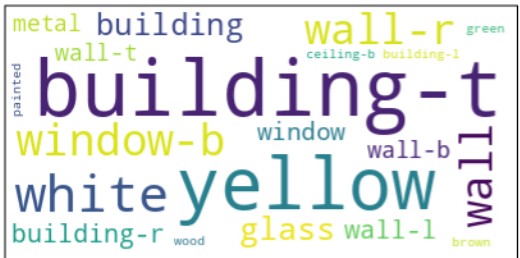

## Lateral

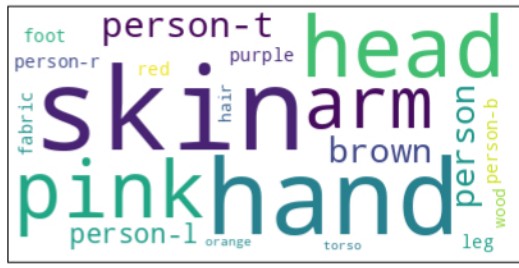

## Parietal

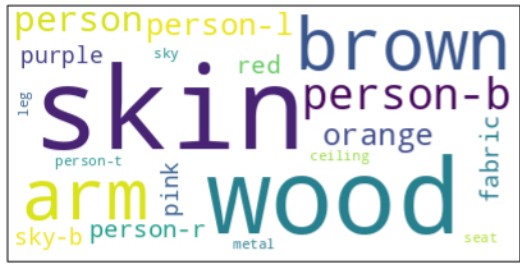

## Medial

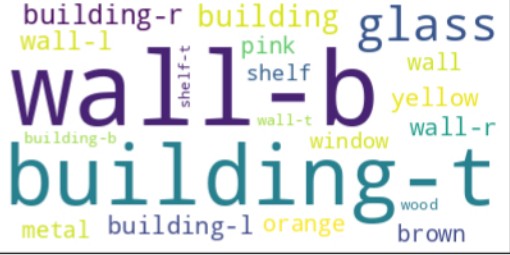

## RSC

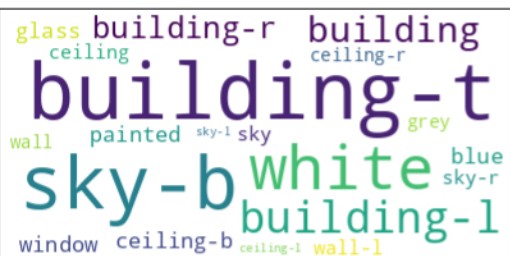

## OPA

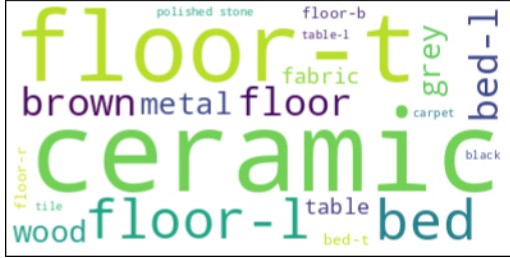

## PPA

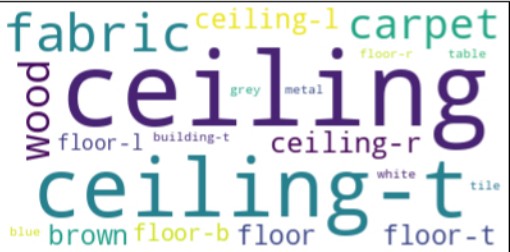

Figure S18: WordCloud for top 20 categories that an ROI selects for in the Places365 dataset. In this visualization, the category size represents the magnitude of the median IOU

