# OpenReview forum: "Brain Dissection: fMRI-trained Networks Reveal Spatial Selectivity in the Processing of Natural Images"
_NeurIPS.cc/2023/Conference — NeurIPS 2023 poster_

### Official Review · Reviewer_MstC · 2023-06-30

**Soundness:** 4 excellent
**Presentation:** 3 good
**Contribution:** 3 good
**Rating:** 6
**Confidence:** 4

**Summary:**

The authors train CNNs to predict the responses of individual voxels across several ROIs to the natural scenes dataset (NSD). They then develop and employ a strategy termed “brain dissection” to uncover the properties/features of images for which specific visual regions are tuned. They highlight gradients within subregions of the visual stream that correspond to tuning variation along mid-level image characteristics including depth, curvature, and object relations.

**Strengths:**

Originality: The methodology, as the authors note, is conceptually similar to Khosla & Wehbe (2022) but is applied towards evaluating sub-region tuning of mid-level visual features rather than reaffirming category selectivity. This is an effective use and reference of an existing computational strategy towards novel scientific investigation. My only comment would be to include some more comprehensive methods and related work description, instead of relying on the reader going back to this original work for the conceptual motivation and validation. A few sentences will suffice.

Quality: This submission is technically sound, uses appropriate methods, and the claims are well supported for the most part. The authors should, however, be a bit more cautious about claims tying the trends they observe in RSC and OPA/PPA to specific functional preferences, e.g., in 4.1.3 and 4.3, as these are somewhat speculative hypotheses that were not explicitly tested, e.g., tying “up” normal to preferences for an “allocentric frame of reference.”

Clarity: The submission is clearly written and well organized.

Significance: The results are important, tie in with existing literature and discussions, and will inform other researchers interested in visual stream organization.

**Weaknesses:**

Wrapped in with above and discussed again below.

**Questions:**

As noted above, I would recommend expanding the discussion of related work and methodology to make for a clearer and more well-packaged reading. It would also be beneficial to tie in more related work supporting or refuting the hypotheses introduced about the subdivision of the “scene network”, such that these data can be appropriately placed into context with other findings and approaches. This limited connection to existing literature, context, and related findings is the work’s greatest weakness.

**Limitations:**

Methodological limitations are mostly well discussed. An additional sentence about limitations with this type of ROI modeling strategy in general, e.g., potential lack of transfer or generalization to real experiments, should be mentioned.

---

> ### Author Rebuttal · Authors · 2023-08-09
>
> **"More comprehensive methods and related work description" "Tie in more related work supporting or refuting the hypotheses"**
>
> Thank you for your feedback on the need for comprehensive methods and an expanded discussion on related works.
>
> - We have included additional motivation points for the method as it relates to previous methods in the global reviewer response. We have added these to the paper.
>
> - In addition, we have added the following paragraphs to the discussion section:
>
> > We observed specific preferences in regions like the RSC, OPA, and PPA. The RSC demonstrated a pronounced preference for greater depths, outdoor object categories, attributes, and relations, and predominantly “right/left” surface normals. On the other hand, the OPA exhibited a preference for proximate depths, intricate 3D geometries, indoor scene object categories, relations, and attributes, along with a higher inclination towards “upward” surface normals. These findings align with existing literature suggesting that OPA is primed for local navigational affordances [1,2], whereas RSC is geared more towards facilitating landmark-based spatial-memory retrieval [3,4]. In the case of the PPA, its preferences spanned a middle ground between the OPA and RSC, albeit leaning slightly more towards the OPA. This supports the notion of PPA's role in encoding scene structure, albeit at a coarser scale than OPA [2]. A salient finding from our study is the pronounced selectivity of RSC for vertical surface normals when contrasted with PPA and OPA, implying its emphasis on encoding vertical structures. In contrast, PPA and OPA demonstrated a marked preference for horizontal supporting structures, such as tabletops and floors. Prior research has highlighted PPA's sensitivity to scene layout and consistent surface arrangement [5]. Our findings further indicate that such selectivity might be particularly driven by these supporting surfaces.
> >
>
> > Further distinguishing between high-level visual ROIs, our study revealed distinct gradients moving from ventro-lateral to medial areas. Ventro-lateral regions exhibited a preference for closer depths, predominantly horizontal surface normals, and darker shading. In contrast, medial areas showed the opposite preferences, which resonates with the idea of distinct processing of foreground objects and distant background elements in these pathways [3]. An added layer of granularity was evident in the observed variability in depth and surface normal selectivity across voxels in the medial and parietal regions. Such variability is indicative of specialized regions tailored for different 3D profiles and global shape processing [6,7]. Furthermore, the parietal region stood out in its pronounced selectivity for spatial relations (like 'on', 'near', etc.), underscoring the significance of spatial relation encoding for this area, a finding that corroborates recent research [7].
>
> **Be more clear about speculative conclusions in the paper.**
>
> We appreciate your feedback. We've refined the paper to clarify speculative statements. For instance, we modified the statement “Surface normals oriented at the camera indicate an egocentric frame of reference..." to "It can be hypothesized, though not yet confirmed, that surface normals facing the camera might suggest an egocentric perspective, while those oriented from the ground could indicate an allocentric viewpoint." All speculative content has been relocated to the discussion section.
>
> **"Limitations with this type of ROI modeling strategy in general"**
>
> Thank you for highlighting the need to discuss the limitations of our ROI modeling strategy in more depth. To expand on some of the limitations of our approach:
>
> 1. While our analysis is primarily restricted to the ROIs hypothesis space, we've supplemented it with additional analyses that encompass entire streams, ensuring a broader scope of interpretation.
>
> 2. We acknowledge that focusing solely on individual voxels might miss out on representations in networks or multiple voxels. This limitation forms the basis for future work, a dissection procedure considering multiple units, not just individual ones.
>
> 3. Our approach relies on pretrained networks to derive spatial measures from images, and this can introduce some estimation errors.
>
> 4. We are also aware that in natural images, certain categories inherently correlate with specific spatial attributes, like scenes predominantly showcasing rectilinear and faraway features, while bodies are usually depicted up close and facing the camera. Our methodology specifically targets ROIs that are widely accepted to encode these spatial properties, which helps to counteract this limitation.
>
> **References**
>
> 1. Lescroart & Gallant (2019). Neuron, 101(1), 178-192.e7.
>
> 2. Bonner & Epstein (2017). Proceedings of the National Academy of Sciences, 114(18), 4793-4798.
>
> 3. Epstein (2008). Trends in Cognitive Sciences, 12, 388–396.
>
> 4. Auger, Mullally, & Maguire (2012). PLoS One, 7, e43620.
>
> 5. Epstein & Kanwisher (1998). Nature, 392(6676), 598-601.
>
> 6. Welchman (2016). Annual Review of Vision Science, 2, 345-376.
>
> 7. Ayzenberg & Behrmann (2022). Journal of Neuroscience, 42(23), 4693-4710.

---

> > ### Comment · Reviewer_MstC · 2023-08-11
> > **Comments addressed**
> >
> > Thanks for dialing back on speculative conclusions and for incorporating more detailed limitations. The improved discussion of related work is a useful addition for paper clarity.

---

### Official Review · Reviewer_9bSv · 2023-07-05

**Soundness:** 3 good
**Presentation:** 3 good
**Contribution:** 3 good
**Rating:** 6
**Confidence:** 5

**Summary:**

Understanding the organization of higher level information representation in the brain is a challenging task in neuroscience. Modern deep learning methods together with big data of brain recording have opened up new opportunities for constructing large-scale models in a data-driven way and for gaining valuable insights about information processing in the brain. The present paper explores this idea by training a deep neural network model for predicting human brain fMRI from natural scene images. By analyzing the feature map generated by the model, the paper reveals spatial and functional organization of the brain for a wide range of high-level visual features.

**Strengths:**

Overall, I find that the paper is well written and that it addresses important questions at the intersection of machine learning and neuroscience.

**Weaknesses:**

 I have a number of concerns about the results and particularly, technical details presented in the paper, which prevented from accepting it directly.

**Questions:**

Major comments
1.	The paper mentions that the model is trained on the NSD dataset including 68400 training images and 3600 validation images. However, the performance of the trained network on estimating fMRI data on the training/validation dataset is missing. Since the main point of the paper is to use a DNN model to gain insights about feature representation in the brain, it is crucial to establish an understanding of how well the model can predict brain activity in the first place. I notice that in the supplementary material, the performance of the model is reported for 3 ROIs (PPA, OPA, RSC) in terms of the Pearson correlation coefficient. However, unless the performance of the model is clearly reported for all ROIs, it is difficult to meaningfully interpret the analysis based on the proposed brain dissection method. For instance, the correlation between the DNN prediction and the ground-truth fMRI data in each ROI could be presented as a flatmap.
2.	One concern regarding the calculation of selectivity (Eq. 1-2) is that a voxel is assigned to a feature value disregarding whether the voxel is truly selective at all. For instance, if L_c masked by M_k across all images is uniformly distributed, then no selectivity can be said about this voxel. To remedy this, the author should rule out these non-selective voxels from the analysis, otherwise it could lead to misleading interpretations of the result.


Minor comments
1.	The dimensionality of the neural network model should be clearly stated for reproducibility.
2.	Since the paper is trained on human fMRI data, a discussion about the biological plausibility of the model would be appropriate. For instance, how much can we infer from the feature representation in the model about that in the real brain? Can we make any experimentally testable predictions?


**Limitations:**

The paper technically sounds. But the technical details are not clear and guarantee reproductivity.

---

> ### Author Rebuttal · Authors · 2023-08-09
>
> **The performance of the trained network on estimating fMRI data is missing.**
>
> Thank you for highlighting the importance of performance reporting for all ROIs. Based on your recommendation, we've provided the Pearson correlation coefficient plotted on a flatmap for all ROis on a held-out test set of NSD data in Figure R1 for example Subject S1. Additionally, we compare our model's mean Pearson correlation to the features from [1] and ImageNet task-optimized features (fit to brain data via Ridge regression) for scene ROIs. Notably, our model surpasses the features from [1] and aligns with AlexNet ImageNet features, even though the AlexNet network has a 76x larger parameter size and is trained on 17.6x more images. We will include correlation flatmaps for all subjects in the revised version of the paper.
>
> **Exclusion of non-selective voxels in analysis**
>
> Thank you for highlighting the potential issue of including non-selective voxels in our analysis. In response to your suggestion:
>
> 1. We've reanalyzed our data by excluding voxels that lack selectivity across all evaluation set images, essentially comparing the distribution of L_c masked by M_k against a uniformly distributed (full image) mask. This exclusion used the Kolmogorov–Smirnov test, with a significance threshold of p>0.01.
>
> 2. Updated findings can be seen in Figure R3 of the provided pdf. This figure presents the absolute depth both with and without the non-selective voxels. Our findings remain stable when excluding non-selective voxels. We'll apply this refined method to all metrics in the final manuscript.
>
> **Clarifying the model's dimensionality for reproducibility**
>
> Thank you for this feedback. To address your concern:
>
> 1. We have updated our paper to include the dimensionality of our network. The dimensionality of the CNN is 784697 and the dimensionality of the transformer network is 7053009.
>
> 2. To ensure reproducibility, we have provided our full code for training and evaluation in the supplementary materials. We commit to open-sourcing our code upon publication. We have also expanded on any missing technical details in the paper.
>
> **Discussing the biological relevance of the model. Can we make any experimentally testable predictions?**
>
> 1. Our method indeed paves the way for several experimentally testable predictions. For example, our findings suggest potential studies of the encoding of outdoor versus indoor scenes, as well as differences in expansive versus enclosed space encoding between RSC, OPA, and PPA. Furthermore, the observed differences in horizontal versus vertical orientation preferences across the scene ROIs might hint at unique encodings, such as those of "supporting" structures or divergences in the reference frame.
>
> 2. It's important to note that our model's architecture need not mirror the brain's intricacies. The primary objective is to accurately predict brain responses. By analyzing our model's activations, we can deduce which image features it is leveraging to make predictions, and thus are most relevant for the responses that emerge in that brain voxel. This ability does not necessarily equate to a biologically accurate architecture, as seen in certain studies like [2].
>
> **References**
>
> 1. Lescroart MD, Gallant JL. Human Scene-Selective Areas Represent 3D Configurations of Surfaces. Neuron. 2019 Jan 2;101(1):178-192.e7. doi: 10.1016/j.neuron.2018.11.004.
>
> 2. St-Yves, G., Allen, E.J., Wu, Y. et al. Brain-optimized deep neural network models of human visual areas learn non-hierarchical representations. Nat Commun 14, 3329 (2023). https://doi.org/10.1038/s41467-023-38674-4

---

> > ### Comment · Reviewer_9bSv · 2023-08-19
> >
> > I appreciate the authors for their response and clarification. I have no further comments.

---

### Official Review · Reviewer_c89B · 2023-07-05

**Soundness:** 3 good
**Presentation:** 4 excellent
**Contribution:** 2 fair
**Rating:** 5
**Confidence:** 4

**Summary:**

This study uses the network dissect method to investigate the feature selectivity of RSC, OFA, and PPA in the human brain. This method is called "brian dissection". In particular, this study focuses on some ecologically important intermediate features, such as depth, surface normals, curvatures, and object relations. Results showed that the three regions show distinct feature preferences.

**Strengths:**

1. To my best knowledge, this is the first study to apply the network dissect method to examine the voxel preferences.
2. The overall method is clear and the presentation is good.

**Weaknesses:**

1. It is unclear to me why this study only focuses on some intermediate features, such as depth, surface normals, curvature. In theory, this method can be also used to study both low-level and high-level visual features.
2. The theoretical contribution is unclear. I agree this study may be the first application of network dissection on voxel preferences. But the idea here is generally incremental. It can certainly obtain some new findings because the method per se is new. But I don't see clear progress being made as compared to previous methods. These types of results may be good for a neuroscience journal. The key point which is missing here is that how these representations are formed.

**Questions:**

My major concern is why this method is better than previous methods. For example, a simple encoding model was developed in ref.[1] and capture several important properties of scene-selective regions. I hope the authors provide evidence why brain dissection is better than this simple encoding-regression model. I understand that brain dissection certainly gives different results and feature maps because they are different methods. But this is not the reason why brain dissection is superior.

Or, different dissection methods could be used and show that the results are highly consistent??


[1] Lescroart MD, Gallant JL. Human Scene-Selective Areas Represent 3D Configurations of Surfaces. Neuron. 2019 Jan 2;101(1):178-192.e7. doi: 10.1016/j.neuron.2018.11.004.

**Limitations:**

1. I trust the results. But the results are completely data-driven. I didn't see how the results add much to our understanding of the function of high-level visual regions.
2. My feeling is that this type of result is suitable for neuroscience journals like Neuroimage. I didn't see any new algorithm-level novelty. It is certainly not interesting to the machine learning community. The results here may be of interests to neuroscientists. I mean the results are valid. But this study only displays some results but does not contribute to how the brain form such representation.

---

> ### Author Rebuttal · Authors · 2023-08-09
>
> **The theoretical contribution is unclear. These types of results may be good for a neuroscience journal. The key point which is missing here is how these representations are formed.**
>
> Thank you for your insightful comments. To enhance the clarity on the theoretical contribution of our paper:
>
> 1. Our work has introduced a novel scale of examining pixel-level spatial feature selectivity during natural image viewing, revealing new insights into representational differences in spatial feature encoding in the visual cortex. For example, our findings suggest a preferential role for PPA and OPA in supporting surfaces. Our findings on depth and surface normal selectivity in the scene and visual stream networks indicate specialized pathways for the processing of scenes with different 3D geometries, such as object spatial relations, flat versus vertical surfaces, and outdoor versus indoor scenes. The implications of these findings range from understanding brain representation modularity to better informing potential advances in representation learning, brain-computer interface applications, and treatments of related disorders.
>
> 2. In comparison to prior studies like [1], our model features align more closely with the mid- and high-level visual cortex representations (Figure R1). We've also introduced a brain dissection method offering pixel-level feature selectivity of spatial measures, a clear advancement over the full-image regression analysis of preceding work. We delve deeper into these distinctions in the global rebuttal response.
>
> We'd also like to emphasize the alignment of our work with NeurIPS's scope:
>
> 1. NeurIPS has historically published neuroscience-centric results. To illustrate this, consider the recent papers: [2,3,4,5].
>
> 2. NeurIPS originated at the crossroads of biological and artificial neural networks. The conference’s Call for Papers explicitly promotes neuroscience findings, referencing its dedication to bridging disciplines such as machine learning and neuroscience.
>
> **Why is this method better than previous methods?**
>
> Addressing your concerns, we have included a comparison of our model to that of the model used in Lescroart & Gallant (2019) [1]. **As shown in Figures R1, our brain dissection model demonstrates a significant improvement (nearly 2x across scene ROIs) in its alignment with the brain responses compared to the baseline features from [1].** Additionally, we've provided a textual explanation of how our brain dissection approach offers more detailed and interpretable insights compared to earlier studies on spatial selectivity in the global reviewer response.
>
> **Different dissection methods could be used and show that the results are highly consistent.**
>
> Thank you for your comment. In response to the suggestion for varied interpretability methods:
>
> 1. We expanded our experiments beyond network dissection. We integrated both gradCAM [6] and raw attention techniques [7]. The former utilizes input features combined with network gradients, while the latter leverages attention scores from transformer architectures.
>
> 2. **Our updated results, detailed in Figure R2, affirm consistency across diverse interpretability methods and network architectures.**
>
> We provide more details in the global reviewer response.
>
> **Why not also focus on low-level and high-level features?**
>
> Indeed, our method can handle features from low- to high-level. We focused on intermediate features like 3D spatial attributes and object relationships because the encodings for intermediate features have proven more challenging to describe compared to well-studied low-level [8,9] and high-level category [10,11] features. Our method enables effective study of these features using natural image data without requiring specialized stimuli or changes to brain imaging techniques. In Section 4.2, we also analyzed high-level features such as object relations, attributes, and categories, at a scale larger than most previous studies (1703 categories, 310 relations, 617 attributes).
>
> **References**
>
> 1. Lescroart & Gallant (2019). Neuron, 101(1), 178-192.e7.
>
> 2. Millet et al. (2022). Advances in Neural Information Processing Systems, 35, 33428-33443.
>
> 3. Wang, A., et al. (2019). Advances in Neural Information Processing Systems, 32.
>
> 4. Khosla et al. (2022). Advances in Neural Information Processing Systems, 35, 9389-9402.
>
> 5. Antonello et al. (2021). Advances in Neural Information Processing Systems, 34, 8332-8344.
>
> 6. Selvaraju et al. (2017). In Proceedings of the IEEE international conference on computer vision, 618-626.
>
> 7. Caron et al. (2021). In Proceedings of the IEEE/CVF international conference on computer vision, 9650-9660.
>
> 8. Carandini et al. (2005). Journal of Neuroscience, 25(46), 10577-10597.
>
> 9. Hubel & Wiesel (1962). The Journal of physiology, 160(1), 106.
>
> 10. Desimone et al. (1984). Journal of Neuroscience, 4(8), 2051-2062.
>
> 11. Grill-Spector & Weiner (2014). Nature Reviews Neuroscience, 15(8), 536-548.

---

> > ### Comment · Reviewer_c89B · 2023-08-19
> >
> > Thanks for your reply. This is an application of ML methods on neuroscience datasets, not an innovation on ML methods itself. It is good to see the consistent results across different methods. This is particularly important because simply applying a known method to a neuroscience dataset reduces the novelty. I raised my score to 5.

---

### Official Review · Reviewer_HWnv · 2023-07-06

**Soundness:** 3 good
**Presentation:** 4 excellent
**Contribution:** 2 fair
**Rating:** 5
**Confidence:** 4

**Summary:**

This paper utilized network dissection model to understand how human brain is functionally mapped to perception of natural scenes. The proposed method is used to examine a range of ecologically important, intermediate properties, including depth, surface normals, curvature, and object relations and find consistent feature selectivity differences.

**Strengths:**

- The paper is well written and easy to understand.
- The paper introduces interesting discussions by employing an AI model to a neuroimaging study.


**Weaknesses:**

- While the paper performs a very interesting experiment from neuroscience perspective, there is not much of a technical contribution. The paper employs an existing model, i.e., network dissection, and does not propose a new or novel approach.
- There is no baseline experiment and there is no way to validate the gain or improvement from the proposed method.
- I think this paper has novelty, but perhaps not in a way that the NeurIPS community expects. At least I would like to see a proposal of a novel approach that lead to novel and improved findings.

**Questions:**

- As far as I know, there are various methods that provide interpretability, e.g., Chefer et al., CVPR 2021. Will the result remain the same if other models are adopted other than the network dissection (which is a bit outdated).
- What would be the feature maps like if some naive models were to be deployed, e.g., MLP or simple CNNs with Class Activation Map?
- Is NSD dataset the only dataset that connects fMRI to natural scene? If so, this would justify experimenting only on a single benchmark regardless of generalization issue.
- What are OPA, RSC, PPA in the abstract? They should be fully spelled out.

**Limitations:**

The authors address limitation of the paper.

---

> ### Author Rebuttal · Authors · 2023-08-09
>
> **Lacking on machine learning technical contribution. Lacking novelty in a way that the NeurIPS community expects.**
>
> We value the feedback provided. To enhance the clarity on the novel and improved findings:
>
> 1. Our work has introduced a novel scale of examining pixel-level spatial feature selectivity during natural image viewing, revealing new insights into representational differences in spatial feature encoding in the visual cortex. For example, our findings suggest a preferential role for PPA and OPA in supporting surfaces. Our findings on depth and surface normal selectivity in the scene and visual stream networks indicate specialized pathways for the processing of scenes with different 3D geometries, such as object spatial relations, flat versus vertical surfaces, and outdoor versus indoor scenes. The implications of these findings range from understanding brain representation modularity to better informing potential advances in representation learning, brain-computer interface applications, and treatments of related disorders.
>
> 2. In comparison to prior studies like [1], our model features align more closely with the mid- and high-level visual cortex representations (Figure R1). We've also introduced a brain dissection method offering pixel-level feature selectivity of spatial measures, a clear advancement over the full-image regression analysis of preceding work. We delve deeper into these distinctions in the global rebuttal response.
>
> We'd also like to emphasize the alignment of our work with NeurIPS's scope:
>
> 1. NeurIPS has historically published neuroscience-centric results. To illustrate this, consider the recent papers: [2,3,4,5].
>
> 2. NeurIPS originated at the crossroads of biological and artificial neural networks. The conference’s Call for Papers explicitly promotes neuroscience findings, referencing its dedication to bridging disciplines such as machine learning and neuroscience.
>
> **"There is no baseline experiment and there is no way to validate the gain or improvement from the proposed method."**
>
> Addressing the baseline concerns, we've compared our method with the model from relevant previous work Lescroart & Gallant (2019) [1]. **As shown in Figures R1, our brain dissection model demonstrates a significant improvement (nearly 2x across scene ROIs) in its alignment with the brain responses compared to the baseline features from [1].** Additionally, we've provided a textual explanation of how our brain dissection approach offers more detailed and interpretable insights compared to earlier studies on spatial selectivity in the global reviewer response.
>
> **"Will the result remain the same if other models are adopted other than the network dissection?"**
>
> Thank you for your insightful comments.
>
> 1. In response to the suggestion for varied interpretability methods, we expanded our experiments beyond network dissection. We integrated both gradCAM [6] and raw attention techniques [7]. The former utilizes input features combined with network gradients, while the latter leverages attention scores from transformer architectures.
>
> 2. **Our updated results, detailed in Figure R2, affirm consistency across diverse interpretability methods and network architectures.**
>
> We provide more details in the global reviewer response.
>
> It's important to mention that we looked into the Chefer et al., CVPR 2021 method you referenced. However, integrating it with our custom architecture posed challenges. Both the GradCAM and raw attention methods we tried, which are baselines in that paper, performed relatively well, and are widely recognized for network interpretability [7,8].
>
> **"What would be the feature maps like if some naive models were to be deployed, e.g., MLP or simple CNNs with Class Activation Map?"**
>
> Our current architecture closely resembles a simple CNN with Class Activation Maps. We also experimented with a transformer architecture and analyzed its attention maps, finding its results consistent with our CNN-based network dissection (Figure R2). More details are available in the global reviewer response.
>
> **"Is NSD dataset the only dataset that connects fMRI to natural scene?"**
>
> The NSD dataset is the only fMRI natural image dataset of its scale, with a total of 73000 COCO images and eight subjects.
>
> **"What are OPA, RSC, PPA in the abstract? They should be fully spelled out."**
>
> We apologize for not expanding the abbreviations of the scene-selective regions. We have updated the abstract with the names fully spelled out: occipital place area (OPA), retrosplenial complex (RSC), and parahippocampal place area (PPA).
>
> **References**
>
> 1. Lescroart & Gallant (2019). Neuron, 101(1), 178-192.e7.
>
> 2. Millet et al. (2022). NuerIPS, 35, 33428-33443.
>
> 3. Wang, A., et al. (2019). NuerIPS, 32.
>
> 4. Khosla et al. (2022). NuerIPS, 35, 9389-9402.
>
> 5. Antonello et al. (2021). NuerIPS, 34, 8332-8344.
>
> 6. Selvaraju et al. (2017). ICCV, 618-626.
>
> 7. Caron et al. (2021). ICCV, 9650-9660.
>
> 8. Linardatos et al. (2020). Entropy, 23(1), 18.

---

> > ### Comment · Reviewer_HWnv · 2023-08-17
> >
> > I appreciate the authors for their clarifications.

---

### Author Rebuttal · Authors · 2023-08-09

We sincerely appreciate the constructive feedback from the reviewers. We are grateful that reviewers HWnv, 9bSv, and MstC acknowledged the clarity and quality of our paper's writing. The novelty of our approach was positively highlighted by HWnv, and the pioneering application of the network dissect method to examine voxel spatial preferences was recognized by c89b. We are pleased that both 9bSv and MstC noted the significance of our work at the intersection of machine learning and neuroscience, emphasizing the depth of insights our research provides into the spatial and functional organization of the brain. The technical soundness of our work was underscored by 9bSv and MstC, with MstC further highlighting its importance in the context of existing literature. We'll address overarching questions here and specific reviewer concerns in their respective responses.

@Reviewers HWnv,c89B: **Lacking on machine learning technical contribution. Mostly relevant for neuroscience.**

1. NeurIPS has historically published neuroscience-centric results, evidenced by recent papers: [1,2,3,4].

2. NeurIPS originated at the intersection of biological and artificial neural networks. Its Call for Papers explicitly promotes neuroscience findings.

3. Our work has introduced a novel scale of examining pixel-level spatial feature selectivity during natural image viewing, revealing novel spatial feature encoding in the visual cortex. For example, our findings suggest a preferential role for PPA and OPA in supporting surfaces. Our findings on spatial selectivity in the scene and visual stream networks indicate specialized pathways for the processing of scenes with different 3D geometries, such as object spatial relations, flat versus vertical surfaces, and outdoor versus indoor scenes. The implications of these findings range from understanding brain representation modularity to better informing potential advances in representation learning, brain-computer interface applications, and treatments of related disorders.

4. In comparison to prior studies like [5], our model features align more closely with the mid- and high-level visual cortex representations (Figure R1). We've also introduced a brain dissection method offering pixel-level feature selectivity of spatial measures, a clear advancement over the full-image regression analysis of preceding work.

@Reviewers HWnv,c89B,9bSv: **Why is brain dissection superior to baseline methods? What is the performance of your model?**

Thank you for your feedback. We recognize the value of having comparative baselines. Addressing the concerns raised:

1. We directly compared our model to the model from Lescroart & Gallant. (2019) [5] on held-out NSD brain data using Pearson correlation. We computed the features in [5] for NSD images using the estimation networks, and fit them to brain data using Ridge regression. **As shown in Figure R1, our brain dissection model demonstrates a significant improvement (nearly 2x across scene ROIs) in its alignment with the brain responses compared to the baseline features from [5].**

2. Contrary to Reviewer c89B, our methodology markedly diverges from Lescroart & Gallant (2019). While they utilize a regression model for whole-image features like depth and surface normals, their approach limits deeper feature selectivity, and adding features requires new regression parameters. In contrast, our brain dissection method provides pixel and region-level selectivity across numerous natural image features, supported by brain-aligned neural networks. **This doesn't just mean different results; it signifies a leap in granularity and interpretability.** For example, where Lescroart & Gallant (2019) identified broad depth variations among OPA, PPA, and RSC, we detail fine spatial selectivity differences across the ROIs.

@Reviewers HWnv,c89B: **Do the results hold with other interpretability methods & network architectures?**

Thank you for your insightful comments. Addressing the concerns raised:

1. In response to the suggestion for varied interpretability methods, we expanded our experiments beyond network dissection. We integrated both gradCAM [6] and raw attention techniques [7]. The former utilizes input features combined with network gradients, while the latter leverages attention scores from transformer architectures. See below for implementation details.

2. **Our updated results, detailed in Figure R2, affirm consistency across diverse interpretability methods and network architectures.** Specifically:
    - The raw attention shows stable mean depth and surface normal selectivity when compared to network dissection. The method replicates significant depth metric increase and a preference for right/left surface normals in RSC compared to PPA and OPA, which prefer lower depths and upright surface normals.
    - GradCAM's depth selectivity aligns with network dissection, showing increased metric depth in RSC versus PPA and OPA. For surface normals, GradCAM mostly reflects prior findings but shows variations in RSC's selectivity. This might be attributed to its adaptation for regression gradients, and occasional inconsistencies as pointed out by Chefer et al. (CVPR 2021).

*Implementation Details*: We combined gradCAM with our initial CNN. For the raw attention method, we opted for the Vision Transformer instead of the paper's CNN. We substituted voxel-specific weights with unique attention heads for each voxel prediction. Attention scores from the [CLS] token in these heads were used to generate voxel-specific attention maps for assessment.

**References**

1. Millet et al. (2022). NeurIPS, 35, 33428-33443.

2. Wang, A., et al. (2019). NeurIPS, 32.

3. Khosla et al. (2022). NeurIPS, 35, 9389-9402.

4. Antonello et al. (2021). NeurIPS, 34, 8332-8344.

5. Lescroart & Gallant. (2019). Neuron, 101(1), 178-192.e7.

6. Selvaraju et al. (2017). ICCV, 618-626.

7. Caron et al. (2021). ICCV, 9650-9660.

---

### Decision · Program_Chairs · 2023-09-21

**Decision:**

Accept (poster)

**Comment:**

The paper introduces the use if network dissect method to examine the voxel preferences.
The methodology, though conceptually similar to previous work (acknowledged in the paper), addresses an novel application consisting in evaluating sub-region tuning of mid-level visual features is novel.
The methodolgy is clear.
The paper is well written and easy to understand.
The claims are well supported for the most part.
The results are important, and will inform other researchers interested in visual stream organization.

The contribution is rather incremental from a  methodological perspective.
There is no baseline experiment and there is no way to validate the gain or improvement from the proposed method.

In short, the contribution is solid and interesting from a neuroscience perspective. As such it deserves publication at NeurIPS.